# Slow and Weak Attractor Computation Embedded in Fast and Strong E-I Balanced Neural Dynamics

**Xiaohan Lin**[1,2]  **Liyuan Li**[3,4]  **Boxin Shi**[1,5]  **Tiejun Huang**[1,5]  **Yuanyuan Mi**[6]  **Si Wu**[2*]

[1] Nat'l Key Lab for Multimedia Information Processing, Nat'l Eng. Lab of Visual Technology,
School of Computer Science, Peking University

[2] Peking-Tsinghua Center for Life Sciences, Academy for Advanced Interdisciplinary Studies
School of Psychological and Cognitive Sciences
Beijing Key Laboratory of Behavior and Mental Health
IDG/McGovern Institute for Brain Research
Center of Quantitative Biology, Peking University

[3] Department of Mathematics, Peking University

[4] Institute of Neuroinformatics, University of Zurich

[5] Institute for Artificial Intelligence, Peking University

[6] Department of Psychology, Tsinghua University

`{lin.xiaohan,liyuanli,shiboxin,tjhuang,siwu}@pku.edu.cn`
`miyuanyuan@tsinghua.edu.cn`

## Abstract

Attractor networks require neuronal connections to be highly structured in order to maintain attractor states that represent information, while excitation and inhibition balanced networks (E-INNs) require neuronal connections to be random and sparse to generate irregular neuronal firing. Despite being regarded as canonical models of neural circuits, both types of networks are usually studied independently, and it remains unclear how they coexist in the brain, given their very different structural demands. In this study, we investigate the compatibility of continuous attractor neural networks (CANNs) and E-INNs. In line with recent experimental data, we find that a neural circuit can exhibit both the traits of CANNs and E-INNs when the neuronal synapses consist of two sets: one set is strong and fast for irregular firing, and the other set is weak and slow for attractor dynamics. In addition, both simulations and theoretical analysis reveal that the network exhibits enhanced performance compared to the case of using only one set of synapses, with accelerated convergence of attractor states and retained E-I balanced condition for localized input. We hope that this study provides insight into how structured neural computations are realized by irregular firings of neurons.

## 1 Introduction

The core of theoretical neuroscience is to understand how different neural circuit models generate different dynamical behaviors, and consequently accomplish different brain functions. Among the proposed models in the literature, continuous attractor neural networks (CANNs) and excitation-inhibition balanced neural networks (E-INNs) are two canonical models that have been widely and successfully applied to describe many brain functions and neuronal response properties. Specifically, CANNs have been applied to explain the representation of continuous features in the brain, including, for instances, orientation tuning in the primary visual cortex [2], action intention in the motor

---

*Corresponding author

37th Conference on Neural Information Processing Systems (NeurIPS 2023).

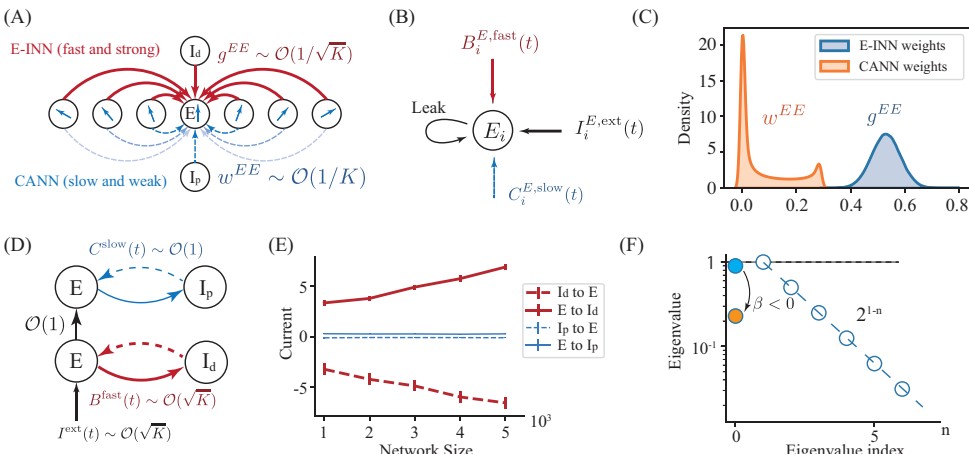

Figure 1: A neural circuit exhibiting traits of both E-INNs and CANNs. (A) E-INN dynamics are mediated by strong, unstructured connections on the order of $\mathcal{O}(1/\sqrt{K})$ and featuring fast synaptic current dynamics (not shown). CANN dynamics rely on weak, translationally invariant connections on the order of $\mathcal{O}(1/K)$ and exhibiting slow synaptic current dynamics (not shown). Only afferent connections to the excitatory neuron are depicted. (B) The kinetics of a single excitatory neuron are influenced by leak current, input currents from E-INN dynamics and CANN dynamics, as well as external input. (C) Weight distribution density plot illustrating the relatively weak nature of CANN weights compared to E-INN weights. $N = 1000$. (D) Intuitive illustration of the coupling mechanism between the two dynamics. Note that the same group of $E$ neurons is shared by both dynamics. They are duplicated in this figure for better visualization. (E) Simulation results demonstrate that E-INN dynamics operate at the scale of $\mathcal{O}(\sqrt{K})$, while CANN dynamics operate at the scale of $\mathcal{O}(1)$. (F) Theoretical analysis of a parallel firing-rate model reveals that E-INN dynamics effectively decrease the first eigenvalue of the CANN dynamics projected onto its dominating motion modes, resulting in faster convergence.

cortex [10], oculomotor signal in the brainstem [25], head direction tuning in the subicular complex [28, 42], place cell tuning in the hippocampus [23, 30], and grid cell tuning in the entorhinal cortex [18, 7, 6]. The structure of a CANN was also directly observed in the neural circuits of *Drosophila* [15, 31] and zebrafish [21]. On the other hand, E-INNs were applied to explain the irregular firing of neurons widely observed in the experiments [26, 32, 33, 5] as well as the fast response property of a neural system to external stimuli [33, 14, 29]. The experimental evidences for the existence of E-I balance in neural systems were also reported [27, 38, 12, 20, 41]. In many modelling studies of large-scale spiking neural networks, E-I balance is imposed as a default condition in the neural dynamics.

Despite being regarded as canonical models of neural circuits, the compatibility of CANNs and E-INNs in the brain remains unclear. In most modeling studies in the literature, the two models are often used separately to account for different brain functions, and there is no formal study on how the two models coexist in a neural circuit. This presents a nontrivial challenge, as the two models exhibit fundamentally different structural characteristics. CANNs rely on highly structured connections with decaying strengths based on neuronal tuning disparity, enabling localized neural activity (bumps) for information representation. In contrast, E-INNs necessitate highly unstructured, random, and sparse connections to ensure statistical independence among input received by individual neurons, facilitating irregular firing patterns. This prompts the question of *how can a neural circuit reconcile these seemingly conflicting requirements?*

In this study, we demonstrate that it is possible for a neural circuit to concurrently incorporate the properties of CANN and E-INN, while also compensating for their respective limitations, through appropriate consideration of connection strengths and synaptic dynamics timescales. Specifically, we consider that the connections are composed of two sets, mediated by two different types of inhibitory neurons respectively (Fig. 1A). One set comprises randomly sparse synapses, characterized by strong synaptic weights scaled with $1/\sqrt{K}$ (where $K$ represents the number of connections) and

fast synaptic dynamics with time constants on the order of 1 ms. The other set consists of synapses which are highly structured as in a CANN, featuring weak synaptic strength that scales with $1/K$ and slow synaptic dynamics with time constants on the order of $100$ ms (Fig. 1C). This model assumption is supported by recent experimental data [24], where strong synapses exhibit no correlation with neuronal selectivity, while the co-activation of weak synapses determines neuronal selectivity.

The mechanism of the neural circuit exhibiting computational traits of CANN and E-INN dynamics simultaneously is illustrated in Fig. 1D-E and can be intuitively understood as follows. E-INN dynamics are mediated by strong synaptic weights that scale with $1/\sqrt{K}$ and fast synaptic dynamics, effectively processing feedforward inputs of $\mathcal{O}(\sqrt{K})$ while always keeping network dynamics in balance; the resulting excitatory and inhibitory synaptic currents, albeit both of $\mathcal{O}(\sqrt{K})$, cancel out and create a constant total synaptic current of $\mathcal{O}(1)$ [33]. This constant total synaptic current serves as the input to CANN dynamics, which operates on a slower timescale with weaker synaptic weights that scale with $1/K$, producing excitatory and inhibitory synaptic current of $\mathcal{O}(1)$. Notably, this study introduces a parallel firing-rate model that accounts for the effect of E-INN dynamics by incorporating a constant term $\beta$ into the CANN connectivity. From the firing-rate model, an analytical solution is derived, allowing us to elucidate the synergistic computation between E-INN and CANN dynamics. We find that E-INN dynamics effectively reduce the eigenvalue of CANN dynamics in the height mode, without affecting other high-order modes (Fig. 1F). We hope that this study can provide insight into how structured neural computations are realized by irregular firings of neurons.

## 2    The Network Model

We present our simulation results using a spiking model. Additionally, we have developed a firing-rate model for theoretical analysis, which is discussed in Sec. 5 in the main text, with a more comprehensive version provided in the Supplementary Information (SI). The spiking model comprises three neuronal populations: an excitatory group (labeled $E$) and two inhibitory groups (shown in Fig. 1A). One group of inhibitory neurons ($I_\mathrm{d}$), presumably formed by somatostatin (SST)-expressing neurons (also known as regular-spiking interneurons), provides distal inhibition to excitatory neurons, resulting in inhibitory currents that interact linearly with the somatic membrane potential. The other group of inhibitory neurons ($I_\mathrm{p}$), presumably formed by parvalbumin (PV)-expressing neurons (also known as fast-spiking interneurons), provides proximal inhibition to excitatory neurons which induces shunting inhibition. The network dynamics depend on two sets of weights: weak weights $w$ from CANN dynamics and strong weights $g$ from E-INN dynamics. The E-INN connectivity pattern is random with connection probability $p$. CANN dynamics, on the other hand, have structured excitatory-to-excitatory connections $w^{EE}$ with translational invariance. Otherwise, the connections are all-to-all.

The dynamics of a neuron is given by

$$\tau_b^m \frac{dV_i^b}{dt} = R_i^b(-g_L V_i^b + I_i^b),\tag{1}$$

where $b = E, I_p, I_d$ denotes the neuron type, $i$ denotes the neuron index, $\tau_b^m$ is the membrane time constant, $g_L$ is the leaky conductance, and $I_i^b$ is the synaptic current received by the neuron. In our experiments, we set membrane resistance $R_i^b = 1$.

For simplicity, we only present the synaptic current received by an $E$ neuron. Refer to SI for the full model. The current consists of three components: a fast recurrent input $B_i^{\mathrm{E,fast}}(t)$, a slow recurrent input $C_i^{\mathrm{E,slow}}(t)$, and an external input $I_i^{\mathrm{E,ext}}(t)$ (Fig 1B):

$$I_i^{\mathrm{E}}(t) = B_i^{\mathrm{E,fast}}(t) + C_i^{\mathrm{E,slow}}(t) + I_i^{\mathrm{E,ext}}(t).\tag{2}$$

The fast recurrent input $B_i^{\mathrm{E,fast}}(t)$ is mediated by a set of strong synaptic connections $g_{i,j}^{\mathrm{E}b}$ that scales with $1/\sqrt{K}$ with fast synaptic current dynamics. $B_i^{\mathrm{E,fast}}(t)$ enables the network to exhibit E-INN dynamics and is given by

$$B_i^{\mathrm{E,fast}}(t) = \Omega^{\mathrm{E}} + \Omega^{\mathrm{I_d}},\tag{3}$$

where $\Omega^{\mathrm{E}} = \sum_j p_{i,j} g_{i,j}^{\mathrm{EE}} f_j^{\mathrm{E}}(t)$ denotes the fast excitatory synaptic input current from population $E$, and $\Omega^{\mathrm{I_d}} = \sum_l p_{i,l} g_{i,l}^{\mathrm{EI_d}} f_l^{\mathrm{I_d}}(t)$ denotes the fast inhibitory synaptic input current from population $I_d$.

$p_{i,j} = \{1, 0\}$ denotes that neurons $i$ and $j$ are connected or unconnected, respectively. $g_{i,j}^{Eb}$ denotes the synaptic strength from neuron $j$ of population $b$ to neuron $i$ in the $E$ population, which is of $\mathcal{O}(1/\sqrt{K})$. $f_j^b(t)$ represents the fast synaptic current input:

$$f_j^b(t) = \sum_k \frac{1}{\tau_{\text{fast}}^s} e^{-(t-t_{j,k})/\tau_{\text{fast}}^s}, \quad b = E, I_{\text{d}}, I_{\text{p}}. \tag{4}$$

Here $t_{j,k}$ denotes the spike time of the $k$th spike of neuron $j$. $\tau_{\text{fast}}^s$ is the fast synaptic time constant. We set $p_{i,j} = 1$ randomly with a probability of 0.25. We choose $\tau_{\text{fast}}^s = 5$ ms such that it's similar to the time constant of AMPA receptors.

The slow recurrent input $C_i^{\text{E,slow}}(t)$ is mediated by a set of weak synaptic connections $w_{i,j}^{Eb}$ that scales with $1/K$ with slow synaptic current dynamics. $C_i^{\text{E,slow}}(t)$ enables the network to exhibit CANN dynamics. It is given by

$$C_i^{\text{E,slow}}(t) = \Gamma^{\text{E}} + \Gamma^{I_{\text{p}}} + \kappa \text{SI}_i(t), \tag{5}$$

where $\Gamma^{\text{E}} = \sum_j w_{i,j}^{\text{EE}} s_j^{\text{E}}(t)$ denotes the slow excitatory synaptic current dynamics from population $E$ and $\Gamma^{I_{\text{p}}} = \sum_l w_{i,l}^{\text{EI}_{\text{p}}} f_l^{I_{\text{p}}}(t)$ denotes the fast inhibitory synaptic current dynamics from population $I_p$. $w_{ij}^{EE} = w_{\text{EE}}^{\max} \exp\left[-\frac{(\theta_i - \theta_j)^2}{2a^2}\right]/N_E$, with $w_{\max}^{\text{EE}}$ being the maximal synaptic weight and $\theta_k \in [-\pi, \pi)$ being the preferred stimulus feature of neuron $k$ (e.g. orientation). $w_{ij}^{EE}$ thus only depends on the tuning disparity $(\theta_i - \theta_j)$ and satisfies translational invariance. $w_{i,l}^{\text{EI}_{\text{p}}}$ represents the connection strength from population $I_p$ to population $E$, which is all-to-all with connection strength scales with $1/K$. $\text{SI}_i(t)$ models the effect of the shunting inhibition as the product of excitatory postsynaptic currents (EPSCs) and inhibitory postsynaptic currents (IPSCs) [13]:

$$\kappa \text{SI}_i(t) = \kappa \left(\Omega^{\text{E}} + \Gamma^{\text{E}} + I_i^{\text{E,ext}}(t)\right) \Gamma^{I_{\text{p}}}. \tag{6}$$

$\kappa$ denotes the strength of shunting inhibition. $s_j^b(t)$ represents slow synaptic current and is given by

$$s_j^b(t) = \sum_k \frac{1}{\tau_{\text{slow}}^s} e^{-(t-t_{j,k})/\tau_{\text{slow}}^s}, \quad b = E. \tag{7}$$

We set the slow synaptic time constant $\tau_{\text{slow}}^s = 150$ ms such that it's similar to the time constant of NMDA receptors. For the full network dynamics, please refer to SI. 1.

## 3 Preservation of E-INN and CANN Dynamics

We simulate the neural circuit with coupled dynamics of E-INNs and CANNs using a spiking model implemented in BrainPy. BrainPy uses JIT to speed up the simulation process [35, 34]. The parameters used in our simulations can be found in SI. 1.

### 3.1 Preservation of E-INN Dynamics

The coupled neural circuit preserves the dynamics and computational properties of E-INNs. Fig. 2A-C demonstrate that the network exhibits irregular activity patterns when the external stimulus is spatially uniform [26, 32]. Additionally, the network's activity level rapidly adjusts to a continuously changing external input, a computational feature exhibited by E-INN dynamics, whereas the counterpart with unbalanced dynamics (where $B_i^{\text{E,fast}}$ is scaled by $1/K$) demonstrates significantly slower adaptation (Fig. 2D) [33].

### 3.2 Preservation of CANN Dynamics

The equilibrium state of CANN dynamics can maintain bump-like persistent activity after the offset of external stimulus, a property thought to be the substrate of working memory. [37, 43]. Moreover, CANN dynamics exhibit neutral stability along its attractor space, endowing a neural circuit the ability to smoothly adapt to a time-varying stimulus [40, 9]. Our model retains these two properties

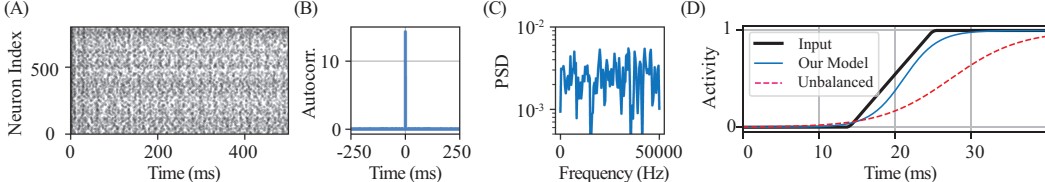

Figure 2: The network preserves E-INN properties. (A) The raster plot of excitatory neurons shows irregular spike patterns, verified by (B) average autocorrelation between neuron pairs, and (C) uniform power spectrum density of network activity. (D) Fitted response curves show the fast response property of the network versus an unbalanced dynamics.

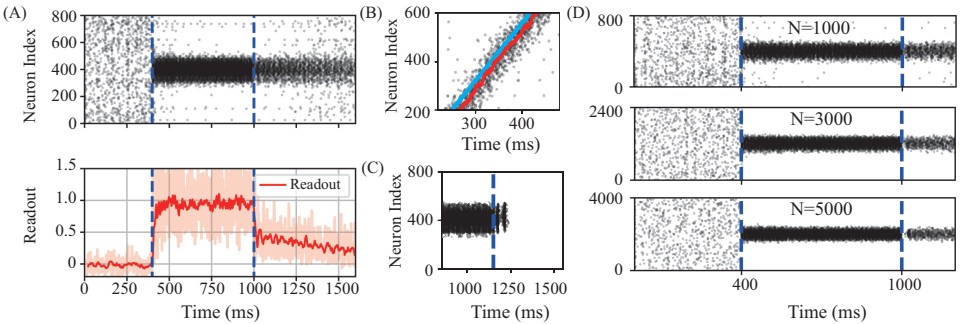

Figure 3: The network preserves CANN dynamics. (A) Top: A raster plot illustrating the network's persistent activity. The blue dashed lines indicate the stimulus onset and offset. Bottom: The network activity projected onto the stimulus, normalized. Shade represents instantaneous readout results. The solid line represents running average calculated over 1.5ms. (B) The network exhibits smooth tracking of a time-varying stimulus. The light blue line represents the stimulus position, while the red line represents the position decoded from the network activity. (C) Fast synaptic dynamics associated with CANN dynamics can lead to synchronization, causing a failure to sustain persistent activity. (D) The model maintains its computational properties across different network sizes using the same parameter set. The vertical dashed lines represent the stimulus onset and offset.

(Fig. 3A-B). Notably, it is crucial for attractor dynamics to be mediated by slow synaptic dynamics. This is because fast synaptic dynamics could lead to synchronization, causing the network to lose persistent activity (Fig. 3C) - a phenomenon referred to as turning-off-with-excitation [16, 11]. This is not a problem for rate-based CANNs but is a biologically realistic issue introduced by spiking dynamics. Moreover, given that the attractor dynamics in our model operates at $\mathcal{O}(1)$ (see Fig. 1D-E), the dynamics can scale with network size without altering parameters (Fig. 3D).

## 4  Synergistic Computation of CANNs and E-INNs

Spiking CANNs and E-INNs have their respective limitations. For instance, a spiking CANN cannot employ fast synaptic current dynamics because of the stability issues (Fig. 3C), while slow synaptic current dynamics severely restricts the neural system's ability to adapt to external changes in a timely manner. On the other hand, E-INN dynamics with global unstructured connectivity cannot maintain balance under localized input [22, 29].

However, we have discovered that CANN and E-INN dynamics not only can coexist and but also exhibit synergistic computation. This coexistence accelerates the convergence speed of CANN dynamics, addressing the limitation of slow synaptic current dynamics while maintaining stability. Additionally, shunting inhibition in the CANN dynamics helps alleviate the issue of imbalance under localized input. We present the simulation results and an intuitive explanation below, and provide a rigorous theoretical analysis in Sec. 5 and SI. 2.

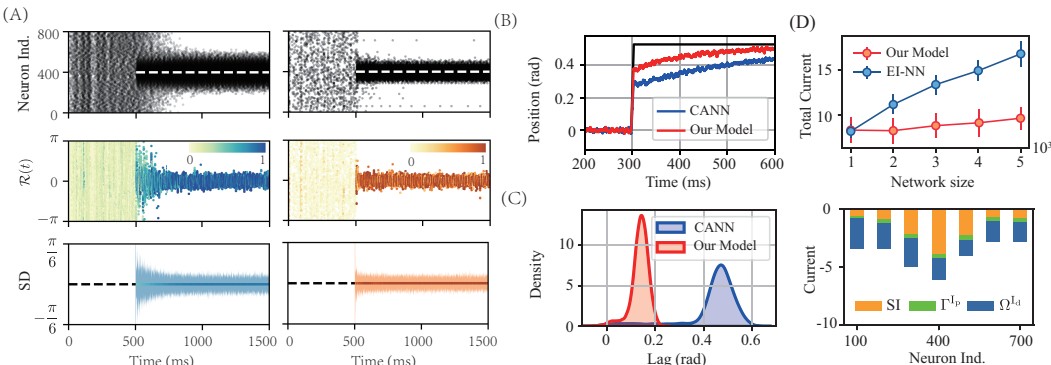

Figure 4: Synergistic computation of CANN and EI-NN dynamics. (A) Decoding results from a CANN (left column) and our model (right column). Top: Raster plots illustrating the activity of excitatory (E) neurons in both models. Middle: Decoded position $\mathcal{R}(t)$ with color indicating the readout confidence $\mathcal{C}(t)$ (see text). Bottom: Standard deviation of decoded position. (B) The network activity (red) in our model adapts more rapidly to sudden changes in the stimulus (black) compared to a CANN with the same parameter settings (blue) (error bars should be included). (Time comparison can be included if feasible). (C) A density plot demonstrates that our model consistently exhibits smaller tracking lag than the CANN model with the same parameter settings. (D) Input current to a center neuron under spatially localized input. Top: Total current increases as the network size increases in E-INN, but remains nearly unchanged in our model. Error bars represent standard deviation over trials. Bottom: Shunting inhibition (orange bar) is strongest at the center neuron with the stimulus in the center and weakest for peripheral neurons.

## 4.1 Synergistic Computation

**Faster population activity readout**  The efficiency of information processing in a neural system critically relies on the speed at which the neural dynamics can be accurately and confidently interpreted. In Fig. 4A, we demonstrate that despite the inherent weakness and slowness of the CANN dynamics in our model, downstream neurons can reliably decode its representation almost immediately after the stimulus onset. In contrast, during the initial phase of stimulus presentation, the original CANN model exhibits more fluctuation and less confidence in the decoding result.

We investigate the population decoding process by calculating the decoded position $\mathcal{R}(t)$ and decoding confidence $\mathcal{C}(t)$. We perform the calculation in complex domain. $\mathcal{R}(t) \in (-\pi, \pi)$ is given by

$$\mathcal{R}(t) = \arg(v(t)), \tag{8}$$

where $v(t) = \sum_j r_j(t)(\cos\theta_j + \mathrm{i}\sin\theta_j)$ is a complex number representing the weighted summation of population activity. $r_j$ represents the firing rate and $\theta_j \in (-\pi, \pi)$ represents the preferred stimulus of neuron $j$. The decoding confidence $\mathcal{C}(t)$ is given by

$$\mathcal{C}(t) = |v(t)| / \max(\tilde{v}_T) \tag{9}$$

which measures the normalized length of the decoded activity vector. $\tilde{v}_T$ represents the smoothed collection of $|v(t)|$ using a time window $T$.

**Faster adaptation to sudden changes**  The rapid adaptation to sudden changes in stimulus is a fundamental aspect of neural computation, as it signifies the neural system's capacity to respond to external fluctuations. In our model, we observe a faster adaptation speed to sudden changes in stimulus compared to the original CANN dynamics under identical parameter settings (Fig. 4B). To quantify this, we measured the time it took for the population activity to reach 95% of the new steady-state value. Remarkably, the unbalanced CANN required almost 50% more time to converge to the new position compared to the balanced CANN. This finding highlights the computational benefits of our model in terms of enhanced responsiveness to dynamic stimuli.

**Smaller tracking lag**  The activity of CANN dynamics can continuously adapt to a time-varying stimulus, such as head movement or external moving targets, with certain amount of lag [40]. The

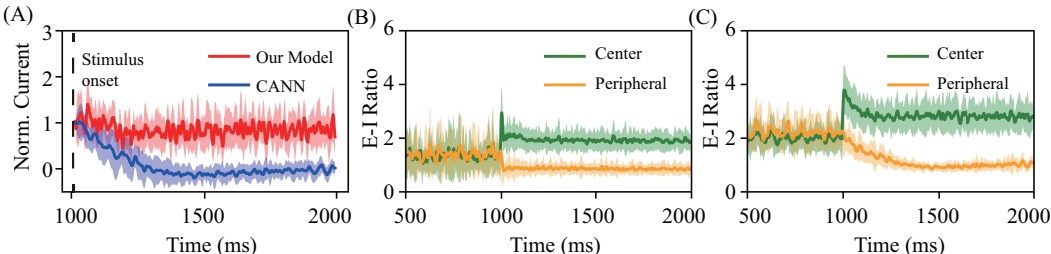

Figure 5: Intuitive explanation of convergence acceleration. Shades represent standard deviation over trials. (A) Visualization of the normalized total input current (see text) to a peripheral neuron in our model and a CANN. (B) E-I ratio in our model. (C) E-I ratio in a CANN.

existence of this lag entails the neural system to come up with solutions to alleviate the delay, for example, by negative feedback [8, 19]. The E-I balanced dynamics also help to mitigate this problem. Fig. 4C shows that our model reliably exhibits smaller tracking lag compared to considering the CANN dynamics alone.

**Maintenance of balanced condition**    One key trait of E-INNs is the production of total input current of $\mathcal{O}(1)$ out of large EPSCs and IPSCs of $\mathcal{O}(\sqrt{K})$, hence the name E-I balance. However, spatially localized stimuli can disrupt this balance due to the lack of spatial structure in the connections between neurons [22, 29]. This means that the total input current to a neuron is no longer of $\mathcal{O}(1)$, but is instead related to its afferent connection number $K$. When the network size $N$ is large, $K = pN$ is also large, which can drive neurons to fire pathologically. In our model, shunting inhibition, mediated through PV neurons, generates a strong inhibitory current that is proportional to the total EPSCs. This greatly mitigates the imbalance issue. Specifically, the neurons that receive the greatest excitatory input also receive the greatest inhibitory input, which keeps the total input to each neuron roughly constant as the network size increases (Fig. 4D). These results are consistent with experimental findings [41], which suggest that PV neurons, but not SST neurons, are essential in adjusting the E-I ratio and maintaining E-I balance.

### 4.2   Intuitive Explanation

To provide an intuitive explanation for the accelerated convergence in our model, we examine the total input current to a neuron located in the peripheral region relative to the stimulus (Fig. 5), as the total input current determines the firing rate of a neuron and its rate of change. To clearly see the evolution trajectory of the total input current, we normalize it using the value at the stimulus onset point. We find that the total current in our model converges more quickly to its equilibrium value without undergoing drastic changes compared to a CANN model, as shown in Fig. 5A. We further examine the EPSCs-to-IPSCs ratio (E-I ratio), as it provides clues on the dynamics of the total input current. The E-I ratio in our model quickly settles to fixed values (Fig. 5B), because the strong E-INN dynamics always keep the network under balance, whereas the E-I ratio in the CANN model slowly converges to its equilibrium value due to the unbalanced nature of the dynamics (Fig. 5C).

## 5   Theoretical Analysis

We present the main results of our theoretical analysis in this section. The full result is in the SI. 2.

In SI 1.1 we present a 1D firing-rate model to investigate the joint dynamics of CANNs and E-INNs. In SI 1.2, we show that this model can be reduced using fast-slow dynamics and mean-field approximation to obtain the following reduced model:

$$\tau_{\text{slow}}^{E} \frac{\partial U_E(x,t)}{\partial t} = -U_E(x,t) + \int_{-\infty}^{\infty} [J(x,x') + \beta]\, r_E(x',t) dx' + I_{\text{ext}}(x,t) \qquad (10)$$

$$r_E(x,t) = \frac{[U_E(x,t)]_+^2}{1 + k \int_{-\infty}^{\infty} [U_E(x',t)]_+^2\, dx'}, \qquad (11)$$

where $U_E(x,t)$ represent the synaptic input to the excitatory neurons at time $t$ for neurons whose preferred stimulus is $x$, $r(x,t)$ represent the firing rate of these neurons, and $I_{\text{ext}}(x,t)$ represent the external inputs. $[.]_+$ denotes the rectifier function and $k$ denotes the strength of global inhibition. The recurrent neural connection matrix $J(x,x') = J_0 \exp\left[-(x-x')^2/(2a)^2\right]$ is translationally invariant, which means that $J(x,x')$ is a function of $(x-x')$.

Importantly, $\beta = \bar{g}^{EE} - \bar{g}^{IE}\bar{g}^{EI}/\bar{g}^{II}$ represents the effect of E-INN dynamics on the CANN dynamics, and $\bar{g}^{ab}$ represents the average connection strength from neural group $b$ to $a$ in the E-INN dynamics. The E-I balanced condition in classical literatures [33] requires that

$$\frac{\bar{g}^{EI}}{\bar{g}^{II}} > \frac{\bar{g}^{EE}}{\bar{g}^{IE}}, \tag{12}$$

which gives $\beta < 0$. Intuitively, the E-INN dynamics affects the CANN dynamics by counteracting the excessive excitatory recurrent inputs through introducing a negative offset to its neural interaction strength. We analyze the case when the magnitude of $\beta$ is sufficiently small for the advantage that it permits an analytical solution of the network stable state for the reduced model. Although there is no theoretical guarantee that the final conclusions of our analysis are applicable to general cases where $|\beta|$ is large, we perform simulations to ensure that for reasonably large $|\beta|$, our conclusion still holds (see Fig. S1 in SI).

We first consider the case when there is no external stimulus. In this case, it is straightforward to check that, for $0 < k < k_c$, the network holds a continuous family of stationary states [39]:

$$\bar{U}_E(x \mid z) = \frac{Ag}{\sqrt{2}} \exp\left[-\frac{(x-z)^2}{4a^2}\right] \tag{13}$$

$$\bar{r}_E(x \mid z) = A \exp\left[-\frac{(x-z)^2}{2a^2}\right] \tag{14}$$

where $z$ represents the center of the activity bump and $A$ is a constant. Previous studies have demonstrated that the CANN dynamics is dominated by a small number of motion modes [40]. Therefore, to simplify the analysis, we can project the network dynamics onto these dominant modes. In this study, we adopt the first two dominant motion modes, which correspond to changes in bump height and position, respectively. We address higher-order motion modes in SI 1.3. The expressions for the first two dominant motion modes are as follows:

$$\phi_0(x \mid z) = \frac{1}{\sqrt{(2\pi)^{1/2}a}} \exp\left[-\frac{(x-z)^2}{4a^2}\right] \tag{15}$$

$$\phi_1(x \mid z) = \frac{(x-z)}{a\sqrt{(2\pi)^{1/2}a}} \exp\left[-\frac{(x-z)^2}{4a^2}\right] \tag{16}$$

The corresponding eigenvalues for the first two motion modes are:

$$\lambda_0 = (1 + \frac{2\sqrt{2\pi}a\beta}{g})(1 - \sqrt{1 - \frac{k}{k_c}}), \text{ and } \lambda_1 = 1. \tag{17}$$

We denote the first eigenvalue when $\beta = 0$ as $\lambda_0^*$, which represents the eigenvalue on the height motion mode without the effects of the E-INN dynamics. For $\beta < 0$, we have $0 < \lambda_0 < \lambda_0^* < 1$, indicating that the CANN will converge faster on the height motion mode when coupled with the E-INN dynamics. The second eigenvalue indicates that the network is still neutrally stable with respect to the second motion mode, i.e. positional shift, and can hold a continuous family of activity bumps. In SI 1.3, we show that $\beta$ does not affect higher-order eigenvalues.

**Faster population activity readout** Given that $0 < \lambda_0 < \lambda_0^* < 1$ when $\beta < 0$, it is evident that the E-INN dynamics facilitates the convergence of the network towards the height direction. Downstream neurons can thus have faster access to the equilibrium state of CANN activity.

**Faster adaptation to sudden changes** Following [9], in SI 1.3 we show that when the external stimulus takes the following form:

$$I_{\text{ext}}(x,t) = \alpha \frac{Ag}{\sqrt{2}} \exp\left[-\frac{(x-z_0)^2}{4a^2}\right] \tag{18}$$

where $\alpha$ is a positive number indicating the input magnitude, the speed of the network's activity adapting to a sudden change (up to the first order of perturbation) is given by:

$$\frac{dz}{dt} = \left\{ \frac{\alpha}{\tau_{\text{slow}}^E} (z_0 - z) \exp\left[ -\frac{(z_0 - z)^2}{8a^2} \right] \right\} R(t)^{-1} \tag{19}$$

where

$$R(t) = 1 + \alpha \int_0^t \frac{dt'}{\tau_{\text{slow}}^E} \exp\left[ -\frac{1 - \lambda_0}{\tau_{\text{slow}}^E} (t - t') - \frac{(z_0 - z(t'))^2}{8a^2} \right] \tag{20}$$

and $z_0$ represents the new input location after the sudden change. Consequently, the following relation holds:

$$\left. \frac{dz}{dt} \right|_{\lambda_0} > \left. \frac{dz}{dt} \right|_{\lambda_0^*}. \tag{21}$$

This implies that coupling CANN dynamics with E-INN dynamics always leads to faster convergence to the new equilibrium location than without E-INN dynamics.

**Smaller tracking lag**  Assuming a constant external stimulus speed $v$, we can substitute $z_0$ with $vt$ in Equation (19), resulting in $s = vt - z$, where $z$ represents the instantaneous activity bump location. Therefore, we have the following equation for the time derivative of the lag distance $s$:

$$\frac{ds}{dt} = v - \frac{\alpha s}{\tau_{\text{slow}}^E} \exp\left( -\frac{s^2}{8a^2} \right) R(t)^{-1} \tag{22}$$

where

$$R(t) = 1 + \alpha \int_{-\infty}^t \frac{dt'}{\tau_{\text{slow}}^E} \exp\left[ -\frac{1 - \lambda_0}{\tau_{\text{slow}}^E} (t - t') - \frac{s(t')^2}{8a^2} \right]. \tag{23}$$

Equilibrium is reached quickly for a sufficiently strong stimulus, with $s(t') = s(t)$ for any $t'$. Using this approximation, we can integrate Equation (23) and obtain:

$$R(t) = 1 + \frac{\alpha}{1 - \lambda_0} \exp\left[ -\frac{s^2}{8a^2} \right] \tag{24}$$

We can then solve for $v$ and obtain:

$$v = \frac{\alpha s}{\tau_{\text{slow}}^E} \exp\left( -\frac{s^2}{8a^2} \right) R(t)^{-1}. \tag{25}$$

Denote the right-hand side as $g(s)$, which is a concave function. The maximum value of $g(s)$ represents the maximal trackable speed $v_{\max}$. Solving $g(s) = v$ for $v < v_{\max}$ gives two solutions: a stable fixed point $s_1 = \bar{s}(v)$ and an unstable fixed point $s_2 = \bar{u}(v)$. The fixed point corresponds to the lag in the smooth tracking task. In SI 1.3, we show that when $\beta < 0$, we have

$$\left. \bar{s}(v) \right|_{\lambda_0} < \left. \bar{s}(v) \right|_{\lambda_0^*}, \text{ and } \max_{\lambda_0} g(s) > \max_{\lambda_0^*} g(s). \tag{26}$$

This implies that the CANN dynamics with E-INN dynamics can tolerate a higher maximal speed than without E-INN dynamics.

## 6  Conclusion and Discussion

In this study, we investigated the coexistence of CANN and E-INN dynamics and found that to successfully fuse these two dynamics, it's crucial to set the strength and speed of their connections appropriately. In line with recent experimental data [24], we suggest making the E-INN connections strong and fast to provide a balanced computational substrate, while keeping the CANN connections weak and slow for stability. Our results demonstrate that the fused dynamics can preserve the unique features of both CANN and E-INN dynamics. Furthermore, we discovered that these two dynamics can work synergistically, allowing for faster downstream readout, faster adaptation to a sudden change stimulus, smaller tracking lag and maintaining balanced condition under localized input. Finally, we developed a firing-rate model to explain this synergy analytically, concluding that the E-INN dynamics essentially decreases the eigenvalue on the height motion mode, leading to faster convergence on this mode and overall dynamics.

Our work builds on two key components: fast E-I balanced dynamics and slow CANN dynamics which have been explored separately by recent studies.

**On fast E-I balanced dynamics** Recent developments in spike-coding networks have demonstrated that coupling a fast E-I balanced dynamics with a slow neural computation dynamics can lead to effective implementation of linear dynamical systems [3, 1]. This idea is further extended by local learning rules based on E-I balanced condition [4] and a minimax optimization objective for various cognitive computations [17]. In these studies, E-I balanced dynamics serve as an efficient and accurate coding strategy for continuous values from discrete spikes using specifically designed connection weights that do not scale with $1/\sqrt{K}$.

**On slow CANN dynamics** Several studies have pointed out that enforcing slow dynamics on CANNs is a necessary choice for biologically realistic model for several reasons. Firstly, fast excitatory dynamics of spiking CANNs can easily lead to turning-off-with-excitation where synchronous firing abruptly shuts down the persistent activity [16, 11]. This is a biologically realistic problem introduced by spiking dynamics. Secondly, persistent activity in CANNs is subject to negative feedback processes such that the bump remains stable. These negative feedback processes (e.g., $GABA_A$-receptor-mediated dynamics) are usually slower than AMPA-receptor-mediated dynamics. As a system with slow negative and fast positive feedback is inherently unstable, NMDA-receptor-mediated dynamics is thus essential for the stability of the dynamical system [36, 37].

Overall, our study sheds light on how structured network can reconcile with the requirement of E-I balanced condition, which entails large and sparse unstructured connection weights. Our findings could offer a possible explanation as to why it's hard to find structured weights in biological circuits in experiments, as a myriad of unstructured neural connections may be required for maintaining a balanced computation environment, and the structured connections that actually implement cognitive tasks are hidden within. We hope this study will inspire further investigation into the complex interplay between different types of connectivity patterns in neural networks.

## Acknowledgement

This work was supported by Science and Technology Innovation 2030-Brain Science and Brain-inspired Intelligence Project (No. 2021ZD0200204, 2021ZD0203700, 2021ZD0203705), National Natural Science Foundation of China (No. 62088102, 62136001, T2122016).

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
