# Slow and Weak Attractor Computation Embedded in Fast and Strong E-I Balanced Neural Dynamics – Supplementary Information

**Xiaohan Lin**[1,2]   **Liyuan Li**[3,4]   **Boxin Shi**[1,5]   **Tiejun Huang**[1,5]   **Yuanyuan Mi**[6]   **Si Wu**[2*]

[1] Nat'l Key Lab for Multimedia Information Processing, Nat'l Eng. Lab of Visual Technology,
School of Computer Science, Peking University

[2] School of Psychological and Cognitive Sciences, IDG/McGovern Institute for Brain Research
Peking-Tsinghua Center for Life Sciences, Center of Quantitative Biology
Academy for Advanced Interdisciplinary Studies, Peking University

[3] Department of Mathematics, Peking University

[4] Institute of Neuroinformatics, University of Zurich

[5] Institute for Artificial Intelligence, Peking University

[6] Department of Psychology, Tsinghua University

{lin.xiaohan,liyuanli,shiboxin,tjhuang,siwu}@pku.edu.cn
miyuanyuan@tsinghua.edu.cn

## 1   The Full Spiking Model

We use the LIF-neuron model given by the equation:

$$\tau_b^m \frac{dV_i^b}{dt} = R_i^b(-g_L V_i^b + I_i^b), \quad b = E, I_{\mathrm{p}}, I_{\mathrm{d}}.$$

In our experiments, we set the membrane resistance $R_i^b = 1$.

The input currents are defined as:

$$I_i^{\mathrm{E}}(t) = B_i^{\mathrm{E,fast}}(t) + C_i^{\mathrm{E,slow}}(t) + I_i^{\mathrm{E,ext}}(t)$$
$$I_i^{\mathrm{I_p}}(t) = C_i^{\mathrm{I_p,slow}}(t)$$
$$I_i^{\mathrm{I_d}}(t) = B_i^{\mathrm{I_d,fast}}(t) + I_i^{\mathrm{I_d,ext}}(t).$$
$$I_i^{b,\mathrm{ext}}(t) = \sqrt{K}\mu_{\mathrm{b}} X_i^{\mathrm{ext}}(t) \quad b = E, I_{\mathrm{d}}$$

Here, $B_i^{b,\mathrm{fast}}(t)$ and $C_i^{b,\mathrm{slow}}(t)$ denote the contributions of fast E-I balanced neural network (E-INN) dynamics and slow continuous attractor neural network (CANN) dynamics to the synaptic input of neuron $i$ in population $b$ at time $t$, respectively. The parameter $\mu_b$ denotes the strength of the external input $X_i^{\mathrm{ext}}(t)$ to population $b$.

**Connectivity Matrix** The conductance parameters that govern the continuous attractor dynamics are denoted by the symbol $w$, with superscripts $a, b \in \{E, I_{\mathrm{p}}\}$. The conductance parameters that govern the E-INN dynamics are denoted by $g$ with superscripts $a, b \in \{E, I_{\mathrm{d}}\}$.

---

[*]Corresponding author

37th Conference on Neural Information Processing Systems (NeurIPS 2023).

$$w_{i,j}^{\mathrm{EE}} = w_{\max}^{\mathrm{EE}} \exp\left[-\frac{(\theta_1 - \theta_2)^2}{2a^2}\right]/N_E$$

$$w_{i,j}^{ab} = w_{\max}^{ab}/(pN_b)$$

$$g_{i,j}^{ab} = g_{\max}^{ab}/\sqrt{pN_b},$$

Remarkably, $w_{i,j}^{ab}$ is scaled by $K$ ($K$ being the number of connections) and $g_{i,j}^{ab}$ is scaled by $\sqrt{K}$. The excitatory-to-excitatory connections involved in continuous attractor dynamics exhibit translational invariance [11], meaning the strength of the connection is solely dependent on the tuning disparity between neurons. Both $I_p$ (PV-expressing) and $I_d$ (SST-expressing) neurons are randomly connected to excitatory neurons [6, 1]. In our study, we consider direct disinhibition within the PV and SST populations, without explicitly accounting for the involvement of vasoactive intestinal polypeptide (VIP)-expressing neurons in disinhibition. Although it is possible to incorporate VIP-mediated disinhibition for biological plausibility, our study does not necessitate its inclusion. Furthermore, consistent with experimental findings, we do not incorporate connections between the PV and SST populations [7].

**Fast Recurrent Input** The fast recurrent input $B_i^{\mathrm{fast}}(t)$ enables the network to exhibit E-INN dynamics. It conforms with classical literature (e.g. [8, 9]). The input to $E$ population is given by:

$$B_i^{\mathrm{E,fast}}(t) = \Omega^{\mathrm{E}} + \Omega^{\mathrm{I_d}}$$

where $\Omega^{\mathrm{E}} = \sum_j p_{i,j} g_{i,j}^{\mathrm{EE}} f_j^{\mathrm{E}}(t)$ and $\Omega^{\mathrm{I_d}} = \sum_l p_{i,l} g_{i,l}^{\mathrm{EI_d}} f_l^{\mathrm{I_d}}(t)$ denote the fast synaptic input current from population $E$ and $I_d$ to population $E$, respectively.

The input to $I_d$ population is given by:

$$B_i^{\mathrm{I_d,fast}}(t) = \Lambda^{\mathrm{E}} + \Lambda^{\mathrm{I_d}}$$

where $\Lambda^{\mathrm{E}} = \sum_j p_{i,j} g_{i,j}^{\mathrm{I_d E}} f_j^{\mathrm{E}}(t)$ and $\Lambda^{\mathrm{I_d}} = \sum_l p_{i,l} g_{i,l}^{\mathrm{I_d I_d}} f_l^{\mathrm{I_d}}(t)$ denote the fast synaptic input current from population $E$ and $I_d$ to population $I_d$, respectively. The conductances $g_{i,j}^{\mathrm{EE}}$, $g_{i,l}^{\mathrm{EI_d}}$, $g_{i,j}^{\mathrm{I_d E}}$, and $g_{i,l}^{\mathrm{I_d I_d}}$ are the corresponding synaptic weights between neurons, and are scaled by $1/\sqrt{K}$ ($K$ being the number of connections). $p_{i,j} = \{1,0\}$ denotes that neurons $i$ and $j$ are connected or unconnected, respectively. We set $p_{i,j} = 1$ randomly with a probability of $0.25$. The fast exponential synaptic current $f^{\mathrm{b}}(t)$ is defined as,

$$f_j^b(t) = \sum_k \frac{1}{\tau_{\mathrm{fast}}^s} e^{-(t-t_{j,k})/\tau_{\mathrm{fast}}^s}, \quad b = E, I_d, I_p.$$

where $\tau_{\mathrm{fast}}^s$ is the time constant for the fast synaptic dynamics, and $t_{j,k}$ denotes the spike time of the $k$th spike of neuron $j$.

**Slow Recurrent Input** The slow recurrent input $C_i^{\mathrm{slow}}(t)$ enables the network to exhibit CANN dynamics:

$$C_i^{\mathrm{E,slow}}(t) = \Gamma^{\mathrm{E}} + \Gamma^{\mathrm{I_p}} + \kappa \mathrm{SI}_i(t)$$

$$C_i^{\mathrm{I_p,fast}}(t) = \Delta^{\mathrm{E}} + \Delta^{\mathrm{I_p}}$$

where $\Gamma^{\mathrm{E}} = \sum_j w_{i,j}^{\mathrm{EE}} s_j^{\mathrm{E}}(t)$ and $\Gamma^{\mathrm{I_p}} = \sum_j w_{i,j}^{\mathrm{EI_p}} f_j^{\mathrm{I_p}}(t)$ denote the input current from population $E$ and $I_p$ to population $E$, respectively. $\Delta^{\mathrm{E}} = \sum_k w_{i,k}^{\mathrm{EI_p}} s_k^{\mathrm{I_p}}(t)$ and $\Delta^{\mathrm{I_p}} = \sum_k w_{i,k}^{\mathrm{I_p I_p}} f_k^{\mathrm{I_p}}(t)$ denote the input current from the population $E$ and $I_p$ to the population $I_p$, respectively. Note that we adopt fast inhibitory synaptic dynamics for the $I_p$ population in the model for better correspondence with our rate-based model in latter sections.

The conductances $w_{i,j}^{\mathrm{EE}}$, $w_{i,j}^{\mathrm{EI_p}}$, $w_{i,k}^{\mathrm{I_p E}}$ and $w_{i,k}^{\mathrm{I_p I_p}}$ are the corresponding synaptic weights between neurons, and are scaled by $1/K$. $\kappa$ represents the strength of shunting inhibition $\mathrm{SI}_i(t)$, which is modeled as the product of excitatory postsynaptic currents (EPSCs) and inhibitory postsynaptic currents (IPSCs) [3]:

$$\kappa \mathrm{SI}_i(t) = \kappa \left(\Omega^{\mathrm{E}} + \Gamma^{\mathrm{E}} + I_i^{\mathrm{E,ext}}(t)\right) \Gamma^{\mathrm{I_p}}.$$

Finally, the slow exponential synaptic current $s^b(t)$ is defined as,

$$s_j^b(t) = \sum_k \frac{1}{\tau_{\text{slow}}^s} e^{-(t-t_{j,k})/\tau_{\text{slow}}^s}$$

where $\tau_{\text{slow}}^s$ is the time constant for the slow synaptic current dynamics.

## 1.1 Parameters

All simulations were conducted with the same set of parameters, unless otherwise specified. Table S1 shows the network parameters used in the simulations. Here, $N_E$, $N_{I_p}$, and $N_{I_d}$ denote the number of excitatory neurons, PV-expressing interneurons, and SST-expressing interneurons, respectively. The total number of neurons in the network is given by $N_E + N_{I_d} + N_{I_p}$. The connection probability between any two neurons is $p$.

Table S1: Network parameter

| $N_E$ | $N_{I_p}$ | $N_{I_d}$ | $N$ | $p$ |
|---|---|---|---|---|
| 800 | 100 | 100 | $N_E + N_{I_d} + N_{I_p}$ | 0.25 |

Table S2 presents the neuron parameters for both excitatory (E) and inhibitory (I) neurons. The parameters include the membrane time constant $\tau_b^m$ for both neuron types, the membrane potential at which the neuron resets $V_{\text{reset}}$, the membrane potential threshold $V_{\text{threshold}}$ that triggers spiking, and the leak conductance $g_L$.

Table S2: Neuron parameters for both E and I neurons

| $\tau_E^m$ | $\tau_I^m$ | $V_{\text{reset}}$ | $V_{\text{threshold}}$ | $g_L$ |
|---|---|---|---|---|
| 20 | 15 | 0. | 1. | 0.15 |

Table S3 shows the synapse parameters used in the simulations. The parameters include the time constants for fast and slow synaptic dynamics. We choose time constants such that $\tau_{\text{fast}}^s$ is similar to the time scale of AMPA receptors, $\tau_{\text{slow}}^s$ is similar to that of NMDA receptors. The parameter $\kappa$ represents the strength of shunting inhibition.

Table S3: Synapse parameters (part 1)

| $\tau_{\text{slow}}^s$ | $\tau_{\text{fast}}^s$ | $k$ |
|---|---|---|
| 150 | 6 | 1 |

Table S4 shows the synapse parameters in the fast synaptic input current dynamics $B_i^{\text{fast}}(t)$ used in the simulations. The parameters include the maximum (measured by magnitude) synaptic conductances $g_{\text{max}}^{EE}$, $g_{\text{max}}^{EI_d}$, $g_{\text{max}}^{I_dE}$, and $g_{\text{max}}^{I_dI_d}$.

Table S4: Synapse parameters (part 2) in the fast synaptic input current dynamics

| $g_{\text{max}}^{EE}$ | $g_{\text{max}}^{EI_d}$ | $g_{\text{max}}^{I_dE}$ | $g_{\text{max}}^{I_dI_d}$ |
|---|---|---|---|
| 7.5 | -14.4 | 15 | -11.4 |

Table S5 displays the synapse parameters in the slow recurrent input dynamics $C_i^{\text{slow}}(t)$ used in the simulations. The parameters include the maximum (measured by magnitude) synaptic weights $w_{\text{max}}^{EE}$, $w_{\text{max}}^{EI_p}$, $w_{\text{max}}^{I_pE}$, and $w_{\text{max}}^{I_pI_p}$.

Lastly, Table S6 presents the input parameters used in the simulations. The input parameters include the feedforward strength $\mu_E$ and $\mu_{I_d}$ to $E$ and $I_d$ populations, respectively. In our experiments, $I_d$

Table S5: Synapse parameters (part 3) in the slow recurrent input dynamics

| $w_{\max}^{\mathrm{EE}}$ | $w_{\max}^{\mathrm{EI_p}}$ | $w_{\max}^{\mathrm{I_pE}}$ | $w_{\max}^{\mathrm{I_pI_p}}$ |
|---|---|---|---|
| 231.25 | -13.75 | 20.0 | -5.0 |

neurons do not receive direct feedforward inputs. This is in line with the typical role of inhibitory neurons in the brain, which primarily serve as interneurons and are not directly targeted by feedforward inputs.

Table S6: Input parameter

| $\mu_{\mathrm{E}}$ | $\mu_{\mathrm{I_d}}$ |
|---|---|
| 0.1 | 0. |

## 2  The Firing-rate Model

We present a firing-rate model to investigate the joint dynamics of continuous attractor neural networks (CANNs) and E-I balanced neural networks (E-INNs). The model builds upon an established rate model for CANNs [12], which we further extend to include a modified rate representation of E-INN dynamics.

### 2.1  Model Description

Denote by $U_E(x,t)$ the synaptic input to the excitatory neurons at time $t$ for neurons whose preferred stimulus is $x$. Let $r(x,t)$ represent the firing rate of these neurons. The firing rate $r(x,t)$ increases with the synaptic input but saturates in the presence of global activity-dependent inhibition. A solvable model capturing these aspects is given by the divisive normalization [10]:

$$r_E(x,t) = \frac{[U_E(x,t)]_+^2}{1 + k \sum_{x'} [U_E(x',t)]_+^2}. \tag{S1}$$

Here, $k$ is a small positive constant controlling the strength of global inhibition. The rectified function is represented by $[.]_+$. The effect of $I_p$ neurons, which provide shunting inhibition in the spiking model, is incorporated in the divisive normalization in Equation (S1).

The dynamics of the synaptic input to the excitatory neurons $U_E(x,t)$ is determined by the external input $I_{\mathrm{ext}}(x,t)$, the synaptic input from the CANN dynamics, the synaptic input from the E-INN dynamics $B(x,t)$, and its own relaxation. It is given by:

$$\tau_{\mathrm{slow}}^E \frac{\partial U_E(x,t)}{\partial t} = -U_E(x,t) + \sum_{x'} J(x,x') r_E(x',t) + I_{\mathrm{ext}}(x,t) + B(x,t). \tag{S2}$$

Here, $J(x,x')$ represents the neural interaction strength from $x'$ to $x$. The key characteristic of CANNs is the translational invariance of their neural interactions. In our model, we choose Gaussian interactions with a range $a$, i.e.

$$J(x,x') = J_0 \exp\left[-\frac{(x-\mathrm{x}')^2}{2a^2}\right].$$

The E-INN dynamics influence the excitatory neurons in the CANN dynamics by providing an input variable $B(x,t)$, whose dynamics is given by:

$$\tau_{\mathrm{fast}}^E \frac{\partial B(x,t)}{\partial t} = -B(x,t) + \sum_{x'} g^{EE}(x,x') r_E(x',t) + \sum_{x'} g^{EI}(x,x') r_I(x',t). \tag{S3}$$

Here, $g^{a,b}(x, x')$ denotes the connectivity strength of neuron $x'$ in population $b$ to neuron $x$ in population $a$ and is given by:

$$g_{ab}^{i,j} = \begin{cases} g_{ab}^{\max}, & p = \alpha \\ 0, & p = 1 - \alpha \end{cases}$$

Since both systems share the same population of excitatory neurons, their firing rates, denoted by $r_E$, are determined by the CANN dynamics in Equation (S1). Additionally, the firing rate of the inhibitory neurons ($I$) is computed by dividing the membrane potential $U_I$ by the threshold $\theta_I$, as the membrane potential essentially integrates the current influx over time.

$$\tau_{\text{fast}}^I \frac{\partial U_I(x, t)}{\partial t} = -U_I(x, t) + \sum_{x'} g^{IE}(x, x') r_E(x', t) + \sum_{x'} g^{II}(x, x') r_I(x', t) \qquad \text{(S4)}$$

$$r_I(x, t) = U_I(x, t) / \theta_I, \qquad \text{(S5)}$$

where $\theta_I$ denotes the threshold of inhibitory neurons. This inhibitory neuronal group $I$ is equivalent to the $I_d$ group in the spiking model. To avoid clutter, we omit the subscript $d$ here since the effect of $I_p$ is absorbed into the divisive normalization, and thus there is not need to distinguish between $I_p$ and $I_d$.

The resulting mathematical formulation is given by:

$$\tau_{\text{slow}}^E \frac{\partial U_E(x, t)}{\partial t} = -U_E(x, t) + \sum_{x'} J(x, x') r_E(x', t) + I_{\text{ext}}(x, t) + B(x, t)$$

$$r_E(x, t) = \frac{[U_E(x, t)]_+^2}{1 + k \sum_{x'} [U_E(x', t)]_+^2}$$

$$\tau_{\text{fast}}^E \frac{\partial B(x, t)}{\partial t} = -B(x, t) + \sum_{x'} g^{EE}(x, x') r_E(x', t) + \sum_{x'} g^{EI}(x, x') r_I(x', t)$$

$$\tau_{\text{fast}}^I \frac{\partial U_I(x, t)}{\partial t} = -U_I(x, t) + \sum_{x'} g^{IE}(x, x') r_E(x', t) + \sum_{x'} g^{II}(x, x') r_I(x', t)$$

$$r_I(x, t) = U_I(x, t) / \theta_I$$

## 2.2 Model Reduction

The model can be reduced under the classical fast/slow dynamical system framework. The E-INN dynamics is a fast dynamics compared to the CANN dynamics. We can thus assume the E-INN dynamics is always at its equilibrium. From Equation (S3) and (S4), we have

$$B(x, t) = \sum_{x'} g^{EE}(x, x') r_E(x', t) + \sum_{x'} g^{EI}(x, x') r_I(x', t)$$

$$U_I(x, t) = \sum_{x'} g^{IE}(x, x') r_E(x', t) + \sum_{x'} g^{II}(x, x') r_I(x', t) \qquad \text{(S6)}$$

Plugging Equation (S5) into Equation (S6), we can rewrite it as,

$$U_I(x, t) = \sum_{x'} g^{IE}(x, x') r_E(x', t) + \frac{1}{\theta_I} \sum_{x'} g^{II}(x, x') U_I(x', t)$$

Using mean-field analysis, we represent the effects of $g^{II}(x, x')$, $g^{II}(x, x')$, $g^{EI}(x, x')$, $g^{IE}(x, x')$ with their mean values, which are denoted as $\bar{g}^{EE}, \bar{g}^{II}, \bar{g}^{EI}, \bar{g}^{IE}$ respectively. Similarly, we have $\bar{U}_I(t) = U_I(x, t) = U(x', t)$. Thus, we can rewrite $B(x, t)$ and $U_I(x, t)$ as

$$B(x, t) = \bar{g}^{EE} \sum_{x'} r_E(x', t) + \bar{g}^{EI} \sum_{x'} r_I(x', t) \qquad \text{(S7)}$$

$$U_I(x, t) = \frac{\bar{g}^{IE} \sum_{x'} r_E(x', t)}{1 - \frac{\bar{g}^{II} N_I}{\theta_I}} = \mu \sum_{x'} r_E(x', t),$$

with

$$\mu = \frac{\bar{g}^{IE}\theta_I}{\theta_I - \bar{g}^{II}N_I}.$$

We can rewrite $B(x,t)$ in Equations (S7) as

$$B(x,t) = \bar{g}^{EE}\sum_{x'} r_E\left(x',t\right) + \frac{\mu \bar{g}^{EI}N_I}{\theta_I}\sum_{x'} \mathrm{r}_E\left(x',t\right)$$
$$= \beta \sum_{x'} r_E(x',t), \tag{S8}$$

where

$$\beta = \bar{g}^{EE} + \frac{\mu}{\theta_I}N_I\bar{g}^{EI}.$$

Finally, we plug Equation (S8) into Equation (S2) and arrive at

$$\tau_{\text{slow}}^{E}\frac{\partial U_E(x,t)}{\partial t} = -U_E(x,t) + \sum_{x'}\left[J\left(x,x'\right) + \beta\right]r_E\left(x',t\right) + I_{\text{ext}}\left(x,t\right). \tag{S9}$$

From Equation (S9), it is straightforward to see the effect of E-INN dynamics is absorbed into the neural interaction strength between neurons in CANN dynamics. When $\beta < 0$, the E-INN dynamics serve to counteract the excitatory recurrent dynamics of CANN.

For the ease of theoretical tractability, we replace the summation with integration and consider $x \in -(\infty,\infty)$. The rationale is to consider a one-dimensional continuous stimulus $x$ encoded by a population of neurons, such as the direction of movement or orientation. And we assume that the range of possible stimulus values is much larger than the range of neuronal interactions, which effectively introduce a neuron sheet with infinite number of neurons. Here, the infinity assumption is merely for the ease of theoretical analysis and the neural system could just use a handful of neurons to perform accurate integration [5]. In our case, we need to rewrite $\beta$ in large $N_I$ limit:

$$\beta = \bar{g}^{EE} - \frac{\bar{g}^{IE}\bar{g}^{EI}}{\bar{g}^{II}}.$$

We note that the $\beta < 0$ condition gives $\bar{g}^{EE}\bar{g}^{II} - \bar{g}^{IE}\bar{g}^{EI} > 0$, which is exactly the requirement for maintaining E-I balance in previous literature [8, 9].

Finally, we pack everything together and arrive at the final reduced model:

$$\tau_{\text{slow}}^{E}\frac{\partial U_E(x,t)}{\partial t} = -U_E(x,t) + \int_{-\infty}^{\infty}\left[J(x,x') + \beta\right]r_E(x',t)dx' + I_{\text{ext}}(x,t)$$
$$r_E(x,t) = \frac{[U_E(x,t)]_+^2}{1 + k\int_{-\infty}^{\infty}[U_E(x',t)]_+^2\,dx'}. \tag{S10}$$

### 2.3 Theoretical Analysis

#### 2.3.1 No External Stimulus

**Equilibrium Solution** We first consider the equilibrium solution when no external stimulus is present i.e. $I_{\text{ext}}(x,t) = 0$. The case when external stimulus is present will be covered in latter sections. We analyze the case when $\beta$ is sufficiently small for the advantage that it permits an analytical solution of the network stable state for the reduced model i.e., Equations (S10). Although there is no theoretical guarantee that the final conclusions of our analysis are applicable to general cases where $\beta$ is large, we perform simulations to ensure that for reasonably large $\beta$, our conclusion still holds (see Fig. S1B). When $\beta$ is sufficiently small, the stable solution is minimally affected by the E-INN. It is straightforward to check that, for $0 < k < k_c$, the network holds a continuous

family of stationary states [11], which can be written as,

$$\bar{U}_E(x \mid z) = \frac{Ag}{\sqrt{2}} \exp\left[-\frac{(x-z)^2}{4a^2}\right] \tag{S11}$$

$$\bar{r}_E(x \mid z) = A \exp\left[-\frac{(x-z)^2}{2a^2}\right]$$

where

$$A = \frac{\left(1 + \sqrt{1 - \frac{8\sqrt{2\pi}ak}{j^2}}\right)}{2\sqrt{2\pi}ak} = \frac{1 + \sqrt{1 - \frac{k}{k_c}}}{2\sqrt{2\pi}ak} \tag{S12}$$

$$k_c = \frac{j^2}{8\sqrt{2\pi}a}$$

$$j = \sqrt{2\pi}aJ_0.$$

In the stable solution, we use the free parameter $z$ to represent the center of the activity profile which has a Gaussian shape.

**Perturbative Analysis**    We next use the perturbative method on the stable state solution (S11)

$$U_E(x,t) = \bar{U}_E(x \mid z) + \delta U_E(x,t).$$

Notice that $z$ is time-independent. Thus when $I_{\text{ext}}(x,t) = 0$ we have

$$\tau_{\text{slow}}^E(t)\frac{\partial}{\partial t}\delta U_E(x,t) = -\delta U_E(x,t) + \int_{-\infty}^{\infty} [J(x,x') + \beta]\,\delta r_E(x',t)\,dx' \tag{S13}$$

We next use $\delta U_E(x,t)$ to rewrite $\delta r_E(x',t)$,

$$\delta r_E(x',t) = \int_{-\infty}^{\infty} \frac{\partial \bar{r}_E(x' \mid z)}{\partial \bar{U}_E(x'' \mid z)}\delta U_E(x'',t)\,dx''$$

$$= \frac{2\bar{U}_E(x' \mid z)}{C}\delta U_E(x',t) - \int_{-\infty}^{\infty} \frac{2k\bar{U}_E(x' \mid z)^2\,\bar{U}_E(x'' \mid z)}{C^2}\delta U_E(x'',t)\,dx'' \tag{S14}$$

where

$$C = 1 + k\int_{-\infty}^{\infty} \bar{U}_E(x \mid z)^2 dx = 1 + \frac{A^2j^2\sqrt{2\pi}ak}{2}.$$

Substituting Equation (S14) into (S13), we get

$$\tau_{\text{slow}}^E(t)\frac{\partial}{\partial t}\delta U_E(x,t) = \int_{-\infty}^{\infty}\int_{-\infty}^{\infty} [J(x,x') + \beta]\left[\frac{2\bar{U}_E(x'' \mid z)}{C}\delta(x'-x'')\right.$$

$$\left. - \frac{2k\bar{U}_E(x' \mid z)^2\,\bar{U}_E(x'' \mid z)}{C^2}\right]\delta U_E(x'',t)\,dx'dx'' - \delta U_E(x,t)$$

Exchanging $x$ and $x'$, we have

$$\tau_{\text{slow}}^E(t)\frac{\partial}{\partial t}\delta U_E(x,t)$$

$$= \int_{-\infty}^{\infty}\int_{-\infty}^{\infty} [J(x,x'') + \beta]\left[\frac{2\bar{U}_E(x' \mid z)}{C}\delta(x''-x')\right.$$

$$\left. - \frac{2k\bar{U}_E(x'' \mid z)^2\,\bar{U}_E(x' \mid z)}{C^2}\right]\delta U_E(x',t)\,dx''dx' - \delta U_E(x,t). \tag{S15}$$

Let

$$G\left(x, x' \mid z\right) = \int_{-\infty}^{\infty} \left[J\left(x, x''\right) + \beta\right] \left[\frac{2\bar{U}_E\left(x' \mid z\right)}{C} \delta\left(x'' - x'\right)\right.$$
$$\left. - \frac{2k\bar{U}_E\left(x'' \mid z\right)^2 \bar{U}_E\left(x' \mid z\right)}{C^2}\right] dx'',$$

Equation (S15) can thus be rewritten in a more compact form

$$\tau_{\text{slow}}^E\left(t\right) \frac{\partial}{\partial t} \delta U_E(x, t) = \int_{-\infty}^{\infty} G\left(x, x' \mid z\right) \delta U_E\left(x', t\right) dx' - \delta U_E(x, t).$$

Hence, $G\left(x, x' \mid z\right)$ can be regarded as the interaction strength from neurons whose preferred stimulus is $x'$ to neurons whose preferred stimulus is $x$. $G\left(x, x' \mid z\right)$ can be written as the sum of two terms, $G^J\left(x, x' \mid z\right)$ and $G^\beta\left(x, x' \mid z\right)$, with

$$G^J\left(x, x' \mid z\right) = \int_{-\infty}^{\infty} J\left(x, x''\right) \left[\frac{2\bar{U}_E\left(x' \mid z\right)}{C} \delta\left(x'' - x'\right) - \right.$$
$$\left. \frac{2k\bar{U}_E\left(x'' \mid z\right)^2 \bar{U}_E\left(x' \mid z\right)}{C^2}\right] dx''$$

$$G^\beta\left(x, x' \mid z\right) = \int_{-\infty}^{\infty} \beta \left[\frac{2\bar{U}_E\left(x' \mid z\right)}{C} \delta\left(x'' - x'\right) - \frac{2k\bar{U}_E\left(x'' \mid z\right)^2 \bar{U}_E\left(x' \mid z\right)}{C^2}\right] dx''.$$

Both $G^J\left(x, x' \mid z\right)$ and $G^\beta\left(x, x' \mid z\right)$ can be integrated explicitly, with

$$G^J\left(x, x' \mid z\right) = \frac{Ag^2}{C\sqrt{\pi}a} \exp\left[-\frac{(x'-z)^2}{4a^2}\right] \exp\left[\frac{(x-x')^2}{2a^2}\right]$$
$$- \frac{kA^3g^4}{2C^2} \exp\left[-\frac{(x-z)^2}{4a^2}\right] \exp\left[-\frac{(x'-z)^2}{4a^2}\right]$$

and

$$G^\beta\left(x, x' \mid z\right) = \frac{\sqrt{2}Ag\beta}{C} \exp\left[-\frac{(x'-z)^2}{4a^2}\right] - \frac{\sqrt{\pi}akA^3g^3\beta}{C^2} \exp\left[-\frac{(x'-z)^2}{4a^2}\right].$$

After gathering and reorganizing terms,

$$G\left(x, x' \mid z\right) = G^J\left(x, x' \mid z\right) + G^\beta\left(x, x' \mid z\right)$$
$$= \frac{Ag^2}{C\sqrt{\pi}a} \exp\left[-\frac{(x'-z)^2}{4a^2}\right] \exp\left[\frac{(x-x')^2}{2a^2}\right]$$
$$- \frac{kA^3g^4}{2C^2} \exp\left[-\frac{(x-z)^2}{4a^2}\right] \exp\left[-\frac{(x'-z)^2}{4a^2}\right]$$
$$+ \frac{\sqrt{2}Ag\beta}{C} \exp\left[-\frac{(x'-z)^2}{4a^2}\right]$$
$$- \frac{\sqrt{\pi}akA^3g^3\beta}{C^2} \exp\left[-\frac{(x'-z)^2}{4a^2}\right]. \tag{S16}$$

To rewrite Equation (S16) into a more compact form, we need some preparation first. From Equation (S12), we can rewrite $C$ as

$$C = \frac{2}{1 - \sqrt{1 - k/k_c}}. \tag{S17}$$

Moreover, from Equation (S1), $C$ can also be expressed as

$$C = \frac{Aj^2}{2}. \tag{S18}$$

Using Equations (S17) and (S18), we arrive at the final expression of $G(x, x' \mid z)$:

$$
\begin{aligned}
G(x, x' \mid z) = {} & \frac{2}{a\sqrt{\pi}} \exp\left[-\frac{(x'-z)^2}{4a^2}\right] \exp\left[\frac{(x-x')^2}{2a^2}\right] \\
& - \frac{1 + \sqrt{1 - \frac{k}{k_c}}}{a\sqrt{2\pi}} \exp\left[-\frac{(x-z)^2}{4a^2}\right] \exp\left[-\frac{(x'-z)^2}{4a^2}\right] \\
& + \frac{2\sqrt{2}\beta}{g} \exp\left[-\frac{(x'-z)^2}{4a^2}\right] \\
& - \frac{\sqrt{2}\beta\left(1 + \sqrt{1 - k/k_c}\right)}{g} \exp\left[-\frac{(x'-z)^2}{4a^2}\right].
\end{aligned}
$$

$G(x, x' \mid z)$ represents the neural interaction from neuron $x$ to $x'$, so it is important to consider its eigenfunctions and eigenvalues. Specifically, the following holds:

- If a certain eigenfunction has an eigenvalue larger than one, then $G(x, x' \mid z)$ is unstable in that direction.
- If a certain eigenfunction has an eigenvalue equal to one, then $G(x, x' \mid z)$ is neutrally stable in that direction.
- If a certain eigenfunction has an eigenvalue less than one, then $G(x, x' \mid z)$ will converge to a stable value as iterations proceed.

**Projection onto QHO Basis**  To determine the eigenfunctions and eigenvalues of $G(x, x' \mid z)$, it is necessary to first select a set of convenient basis functions. Previous studies [11, 2] have suggested that the quantum harmonic oscillator (QHO) can be used as a basis function for this purpose. The QHO is a set of complete and orthogonal basis functions in the $L^2$-space, and is given by the following expression:

$$\phi_n(x \mid z) = \frac{1}{Z_n} \exp\left[-\frac{(x-z)^2}{4a^2}\right] H_n\left(\frac{x-z}{\sqrt{2}a}\right).$$

Here, $H_n$ denotes the $n$-th term of the Hermitian polynomial, while $Z_n$ is a normalization constant that can be obtained by evaluating $\int |\phi_n(x|z)|^2 = 1$. The first four terms of QHOs correspond to distortions in the direction of height, positional shift, width, and skewness. Therefore, the use of the QHOs as the basis function can significantly enhance the theoretical analysis of the system [11, 2].

To project $G(x, x'|z)$ onto QHO basis, we denote

$$G_{mn} = \int\int \phi_m(x|z) G(x, x'|z) \phi_n(x'|z) dx dx'.$$

By integrating over the range of $x$ and $x'$, the matrix elements $G_{mn}$ give the strength of the interaction between the $m$th and $n$th basis functions. The projection of the interaction kernel onto both basis functions is necessary to calculate the coupling between them accurately and determine the overall behavior of the system.

**Eigendecomposition of** $G_{mn}$  To calculate the eigenvalues and eigenvectors of $G_{mn}$, we first write out the general form of the QHO using the Rodrigues formula

$$\phi_n(x \mid z) = \frac{(-1)^n(\sqrt{2}a)^{n-1/2}}{\sqrt{\pi^{1/2}n!2^n}} \exp\left[\frac{(x-z)^2}{4a^2}\right] \left(\frac{d}{dx}\right)^n \exp\left[-\frac{(x-z)^2}{2a^2}\right]. \tag{S19}$$

Let

$$f_n^J(x \mid z) = \int_{-\infty}^{\infty} G^J(x, x' \mid z)\, \phi_n(x' \mid z)\, dx' \tag{S20}$$

$$f_n^\beta(x \mid z) = \int_{-\infty}^{\infty} G^\beta(x, x' \mid z)\, \phi_n(x' \mid z)\, dx' \tag{S21}$$

$$f_n(x \mid z) = f_n^J(x \mid z) + f_n^\beta(x \mid z) \tag{S22}$$

Using Equation (S19), we have

$$f_n^J(x \mid z)$$

$$= \frac{(-1)^n 2(\sqrt{2}a)^{n-\frac{1}{2}}}{a\sqrt{\pi^{3/2} n! 2^n}} \int_{-\infty}^{\infty} dx' \exp\left[-\frac{(x-x')^2}{2a^2}\right] \left(\frac{d}{dx'}\right)^n \exp\left[-\frac{(x'-z)^2}{2a^2}\right]$$

$$- \frac{\left(1+\sqrt{1-k/k_c}\right)(-1)^n(\sqrt{2}a)^{n-\frac{1}{2}}}{a\sqrt{\pi^{3/2} n! 2^{n+1}}} \exp\left[-\frac{(x-z)^2}{4a^2}\right] \int_{-\infty}^{\infty} dx' \left(\frac{d}{dx'}\right)^n \exp\left[-\frac{(x'-z)^2}{2a^2}\right]$$

$$f_n^\beta(x \mid z) = \frac{(-1)^n 2\beta(\sqrt{2}a)^{n-\frac{1}{2}}}{g\sqrt{\pi^{1/2} n! 2^{n-1}}} \int_{-\infty}^{\infty} dx' \left(\frac{d}{dx'}\right)^n \exp\left[-\frac{(x'-z)^2}{2a^2}\right]$$

$$- \frac{\beta\left(1+\sqrt{1-k/k_c}\right)(-1)^n(\sqrt{2}a)^{n-\frac{1}{2}}}{g\sqrt{\pi^{1/2} n! 2^{n-1}}} \int_{-\infty}^{\infty} dx' \left(\frac{d}{dx'}\right)^n \exp\left[-\frac{(x'-z)^2}{2a^2}\right]$$

Notice that when $n \geq 1$, we have

$$\int_{-\infty}^{\infty} dx' \left(\frac{d}{dx'}\right)^n \exp\left[-\frac{(x'-z)^2}{2a^2}\right] = 0$$

Thus when $n = 0$

$$f_0(x \mid z) = \left(1 - \sqrt{1-\frac{k}{k_c}}\right)\phi_0(x \mid z) + \frac{\left(1-\sqrt{1-\frac{k}{k_c}}\right)2\sqrt{\pi}a\beta}{g\sqrt{(2\pi)^{1/2}a}}, \tag{S23}$$

and when $n \geq 1$, $g_n^\beta(x \mid z)$ has no effect and hence

$$f_n(x \mid z) = \frac{(-1)^n 2(\sqrt{2}a)^{n-\frac{1}{2}}}{a\sqrt{\pi^{3/2} n! 2^n}} \int_{-\infty}^{\infty} dx' \exp\left[-\frac{(x-x')^2}{2a^2}\right] \left(\frac{d}{dx'}\right)^n \exp\left[-\frac{(x'-z)^2}{2a^2}\right] \quad (n \geq 1).$$

We next integrate $f_n(x \mid z)$ for $n \geq 1$ explicitly and obtain

$$f_n(x \mid z) = \frac{(-1)^n 2(\sqrt{2}a)^{n-\frac{1}{2}}}{\sqrt{\pi^{1/2} n! 2^n}} \left(\frac{d}{dx}\right)^n \exp\left[-\frac{(x-z)^2}{4a^2}\right].$$

Given the definition in Equations (S20), (S21) and (S22), we can derive $G_{mn}$ using

$$G_{mn} = \int_{-\infty}^{\infty} \phi_m(x \mid z) f_n(x \mid z)\, dx.$$

Specifically, when $m = n = 0$, from Equation (S23) we can integrate $G_{00}$ explicitly:

$$G_{00} = \left(1 + \frac{2\sqrt{2\pi}a\beta}{g}\right)\left(1 - \sqrt{1-\frac{k}{k_c}}\right).$$

For other $m$ and $n$ values we follow [2]:

$$G_{mn} = 2^{1-n}\sqrt{\frac{n!}{m!}} \oint \frac{dt}{2\pi i\, t^{n-m+1}} \exp\left(-\frac{t^2}{2}\right) \quad \text{(when } n \geq 1),$$

where $t$ is a complex variable. This integral can be evaluated using the residue theorem from complex analysis, which allows us to express the integral as a sum over the singularities of the integrand in the complex plane. The singularities of the intergrand $\exp(-t^2/2)/t^{n-m+1}$ are the points where the denominator $t^{n-m+1}$ is zero, which occur at $t = 0$. Thus, we can use the Laurent expansion to evaluate the contour integral.

The final result is,

$$G_{mn} = \begin{cases} (1 + \frac{2\sqrt{2\pi}a\beta}{g})(1 - \sqrt{1 - \frac{k}{k_c}}), & m = n = 0; \\ 2^{1-n}\sqrt{\frac{n!}{m!}}\frac{(-1)^{\frac{n-m}{2}}}{2^{\frac{n-m}{2}}(\frac{n-m}{2})!}, & n - m \text{ being an even integer}; \\ 0, & \text{otherwise.} \end{cases}$$

The elements of the matrix $\mathbf{G}$ are:

$$\mathbf{G} = \begin{pmatrix} \left(1 + \frac{2\sqrt{2\pi}a\beta}{g}\right)\left(1 - \sqrt{1 - \frac{k}{k_c}}\right) & 0 & -\frac{\sqrt{2}}{4} & 0 & \cdots \\ 0 & 1 & 0 & -\frac{\sqrt{6}}{8} & \cdots \\ 0 & 0 & \frac{1}{2} & 0 & \cdots \\ 0 & 0 & 0 & \frac{1}{4} & \cdots \\ \cdots & & \cdots & \cdots & \cdots \end{pmatrix}.$$

Notice that matrix $\mathbf{G}$ is an upper triangular matrix. We can read its eigenvalues directly from its diagonal elements. Hence, expressed in the QHO basis, the eigenvalues of the kernel $G(x, x' \mid z)$ are,

$$\lambda_0 = \left(1 + \frac{2\sqrt{2\pi}a\beta}{g}\right)\left(1 - \sqrt{1 - \frac{k}{k_c}}\right)$$

$$\lambda_n = 2^{1-n}, \text{ for } n \geq 1.$$

Computing the first two eigenvectors of the kernel $G(x, x' \mid z)$ is straightforward since there are only one nonzero elements in the each of the first two columns of $\mathbf{G}$:

$$\mathbf{v}_0 = (1, 0, 0, \cdots)^T$$
$$\mathbf{v}_1 = (0, 1, 0, \cdots)^T.$$

Consequently, the first two right eigenfunctions of the kernel $G(x, x' \mid z)$ are,

$$u_0^R(x \mid z) = \sum_l v_{0,l}\phi_l(x \mid z) = \phi_0(x \mid z)$$

$$u_1^R(x \mid z) = \sum_l v_{1,l}\phi_l(x \mid z) = \phi_1(x \mid z).$$

That is, the first two eigenfunctions of the kernel $G(x, x' \mid z)$ are precisely the first two basis functions of QHO. However, the remaining eigenfunctions are more intricate and are not relevant to our present investigation, and hence are omitted in this study.

**Implications** The first two eigenfunctions of the kernel $G(x, x' \mid z)$ are also the first two basis functions in QHO, which describe distortions in the height and positional shift modes.

Let us define

$$\lambda_0^* \equiv \lambda_0\Big|_{\beta=0} = \left(1 - \sqrt{1 - \frac{k}{k_c}}\right).$$

$\lambda_0^*$ thus corresponds to the first eigenvalue when the effect of E-INN dynamics is neglected. When $\beta < 0$, we have

$$\lambda_0 = \left(1 + \frac{2\sqrt{2\pi}a\beta}{g}\right)\left(1 - \sqrt{1 - \frac{k}{k_c}}\right) < \lambda_0^*$$

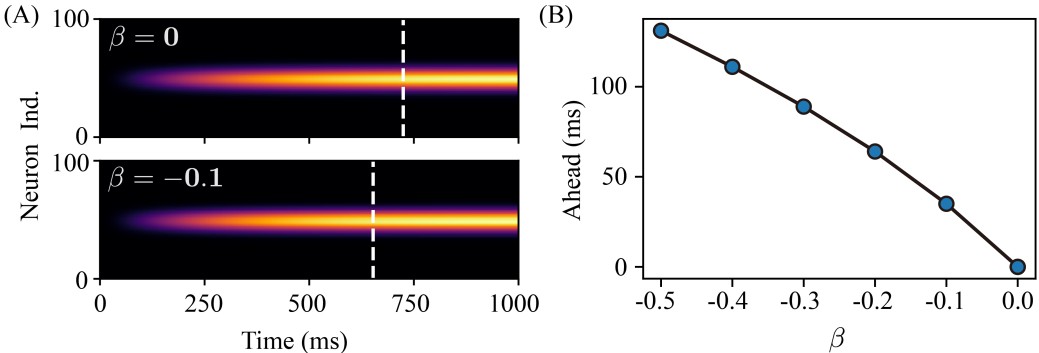

Figure S1: Simulation results on the reduced firing-rate model. (A) The firing-rate model converges faster when $\beta < 0$. White dashed lines indicate when the bump height is 95% of its final value. (B) The convergence leading time vs. different $\beta$ values.

and

$$\lambda_1 = 1.$$

Therefore, distortions in the height mode will decay more rapidly in the presence of E-INN dynamics than in its absence (Fig. S1).

The second eigenvalue indicates that the coupled network dynamics is neutrally stable in the second motion mode. Any distortions in the positional shift mode will remain unchanged. This means that the ability to maintain a continuous family of activity bumps, as in the original CANN model, is not compromised when E-INN dynamics is included.

In our case, the E-INN dynamics in the rate-based model has no effect on higher motion modes, since $\beta$ does not appear in the eigenvalues other than the first one. Consequently, distortions in these modes will decay at the same rate regardless of whether E-INN dynamics are coupled. Nonetheless, these higher order modes decay exponentially fast due to eigenvalues $\lambda_n = 2^{1-n}$ when $n \geq 1$. Therefore, these modes do not dominate the convergence rate of CANN dynamics.

### 2.3.2 Suddenly Moving Stimulus

Section 2.3.1 considers the convergence rate of the model near the equilibrium state when no stimulus is present. We deal with the scenario when the external stimulus is present in this section. Specifically, we investigate how E-INN dynamics contribute to the behavior of the model under a suddenly moved stimulus in this section.

Assuming the presence of a constant external stimulus $I_{\text{ext}}(x, t)$, we consider the model reaching its equilibrium state for $t < 0$. At time $t = 0$, the external stimulus suddenly moves to a nearby position $z_0$.

Without loss of generality, we assume that for $t > 0$, the external stimulus takes the form:

$$I_{\text{ext}}(x, t) = \alpha \frac{Ag}{\sqrt{2}} \exp\left[-\frac{(x - z_0)^2}{4a^2}\right],$$

where $\alpha$ is a positive number.

As the distortion $\delta U_E(x, t)$ is primarily composed of positional shift, with high-order shape distortions from other motion modes, we can assume the solution for $U_E(x, t)$ to be of the form:

$$U_E(x, t) = \bar{U}_E(x \mid z(t)) + \delta U_E(x, t)$$

$$= \bar{U}_E(x \mid z(t)) + \sum_{n=0}^{\infty} a_n(t)\phi_n(x \mid z(t)) \tag{S24}$$

The external input can also be expressed in a similar way:

$$I_{\text{ext}(x,t)} = \sum_{n=0}^{\infty} I_n(t)\phi_n(x \mid z),$$

where $I_n = \int I_{\text{ext}}(x,t)\phi_n(x \mid z)dx$ is the projection of the external input onto the $n$th basis function.

Following [2], the coefficient $a_n(t)$ in Equation (S24) can be expressed as:

$$\left(\frac{d}{dt} + \frac{1-\lambda_n}{\tau_{\text{slow}}^E}\right) a_n = \frac{I_n}{\tau_{\text{slow}}^E} - \left[U_0\sqrt{(2\pi)^{1/2}a}\delta_{n1} + \sqrt{n}a_{n-1} - \sqrt{n+1}a_{n+1}\right]\frac{1}{2a}\frac{dz}{dt}$$

$$+ \frac{1}{\tau_{\text{slow}}^E}\sum_{r=1}^{\infty}\sqrt{\frac{(n+2r)!}{n!}}\frac{(-1)^r}{2^{n+3r-1}r!}a_{n+2r}. \tag{S25}$$

Using the center-of-mass definition:

$$z(t) = \frac{\int_{-\infty}^{\infty} dx U(x,t)x}{\int_{-\infty}^{\infty} dx U(x,t)}$$

we obtain:

$$\frac{dz}{dt} = \frac{2a}{\tau_{\text{slow}}^E}\frac{I_1 + \sum_{n=3,\text{ odd}}^{\infty}\sqrt{\frac{n!!}{(n-1)!!}}I_n + a_1}{U_0\sqrt{(2\pi)^{1/2}a} + \sum_{n=0,\text{ even}}^{\infty}\sqrt{\frac{(n-1)!!}{n!!}}a_n}. \tag{S26}$$

Equations (S25) and (S26) are master equations of the perturbative analysis.

For our purposes, in the following we will only consider coefficient $a_0$ and set high-order coefficient $a_n$ to 0 for $n \geq 1$, as the E-INN dynamics does not have effect on high-order motion modes. We will consider $I_n$ up to $n = 1$, and set $I_n$ to 0 for $n \geq 2$, as $I_1$ drives the positional shift of the network dynamics, and thus cannot be neglected.

From Equations (S25) and (S26) we obtain:

$$\left(\frac{d}{dt} + \frac{1-\lambda_0}{\tau_{\text{slow}}^E}\right) a_0 = \frac{\alpha Ag}{\sqrt{2}\tau_{\text{slow}}^E}\sqrt{(2\pi)^{1/2}a}\exp\left[-\frac{(z_0-z)^2}{8a^2}\right] \tag{S27}$$

and

$$\frac{dz}{dt} = \frac{2a}{\tau_{\text{slow}}^E}\frac{I_1}{Ag\sqrt{(\pi/2)^{1/2}a} + a_0}$$

$$= \frac{\alpha Ag\sqrt{(2\pi)^{1/2}a}(z_0-z)\exp\left[-\frac{(z_0-z)^2}{8a^2}\right]}{\tau_{\text{slow}}^E\left(Ag\sqrt{(2\pi)^{1/2}a} + \sqrt{2}a_0\right)} \tag{S28}$$

It is noteworthy that $I_1$ and $a_0$ are functions of $t$, but for the sake of brevity, we shall drop the dependence. By employing the solution to $\dot{x} = ax + g(t)$, which is given by $x(t) = e^{at}\int_0^t e^{-as}g(s)ds + c$, and taking into account the boundary condition that the network dynamics is in its equilibrium state when $t = 0$ (i.e., $a_0 = 0$), we can derive the solution to Equation (S27):

$$a_0(t) = \int_0^t \frac{dt'}{\tau_{\text{slow}}^E}\exp\left[-\frac{1-\lambda_0}{\tau_{\text{slow}}^E}(t-t')\right]\left\{\alpha\frac{Ag}{\sqrt{2}}\sqrt{(2\pi)^{1/2}a}\exp\left[-\frac{(z_0-z)^2}{8a^2}\right]\right\}. \tag{S29}$$

Combining Equations (S28) and (S29) we get:

$$\frac{dz}{dt} = \left\{\frac{\alpha}{\tau_{\text{slow}}^E}(z_0-z)\exp\left[-\frac{(z_0-z)^2}{8a^2}\right]\right\}R(t)^{-1} \tag{S30}$$

where

$$R(t) = 1 + \alpha \int_0^t \frac{dt'}{\tau_{\text{slow}}^E} \exp\left[-\frac{1-\lambda_0}{\tau_{\text{slow}}^E}(t-t') - \frac{(z_0 - z(t'))^2}{8a^2}\right]. \tag{S31}$$

Equations (S30) and (S31) provide insight into the speed of network adaptation to a suddenly moving stimulus when accounting for the effects of changes in bump height on the network dynamics. In Section 2.3.1, we introduced $\lambda_0^*$ as the first eigenvalue for CANN without E-INN dynamics:

$$\lambda_0^* = \left(1 - \sqrt{1 - \frac{k}{k_c}}\right).$$

For $\beta < 0$, we obtain

$$\lambda_0 = \left(1 + \frac{2\sqrt{2\pi}a\beta}{g}\right)\left(1 - \sqrt{1 - \frac{k}{k_c}}\right) < \lambda_0^*.$$

Thus, we have the relationship

$$\left.\frac{dz}{dt}\right|_{\lambda_0} > \left.\frac{dz}{dt}\right|_{\lambda_0^*}$$

holding for arbitrary $z$ values. That is, when the stimulus suddenly moves to a new location, the coupled network dynamics always converge faster to the new location.

### 2.3.3 Smoothly Moving Stimulus

In this section, we examine the tracking dynamics of a smoothly moving stimulus. A CANN possesses the ability to track a smoothly moving stimulus up to a maximum speed, denoted as $v_{\text{max}}$. In the absence of negative feedbacks [4], the activity bump position consistently lags behind the true position of the stimulus, and we denote the magnitude of this lag as $s$.

Assuming that the external stimulus is moving at a constant speed $v$, we can substitute $z_0$ with $vt$ in Equation (S30), resulting in $s = vt - z$, where $z$ represents the instantaneous location of the activity bump.

To investigate the effect of E-INN dynamics on the smooth tracking ability of CANNs, we examine the changes in the lag distance $s$ and the maximal trackable speed $v_{\text{max}}$.

By substituting $s = vt - z$ into Equation (S30), we obtain the following expression for the time derivative of the lag distance $s$:

$$\frac{ds}{dt} = v - \frac{\alpha s}{\tau_{\text{slow}}^E} \exp\left(-\frac{s^2}{8a^2}\right) R(t)^{-1}$$

where

$$R(t) = 1 + \alpha \int_{-\infty}^t \frac{dt'}{\tau_{\text{slow}}^E} \exp\left[-\frac{1-\lambda_0}{\tau_{\text{slow}}^E}(t-t') - \frac{s(t')^2}{8a^2}\right]. \tag{S32}$$

At equilibrium, we have $s(t') = s(t)$ for any $t'$. For a sufficiently strong stimulus, this equilibrium is reached quickly. By using this approximation, we can integrate Equation (S32) explicitly and obtain:

$$R(t) = 1 + \frac{\alpha}{1-\lambda_0} \exp\left[-\frac{s^2}{8a^2}\right]$$

We can then solve for $v$ and obtain:

$$v = \frac{\alpha s}{\tau_{\text{slow}}^E} \exp\left(-\frac{s^2}{8a^2}\right) R(t)^{-1}.$$

Let us define the function $g(s)$ as::

$$g(s) = \frac{\alpha s}{\tau_{\text{slow}}^E} \exp\left(-\frac{s^2}{8a^2}\right) R(t)^{-1}.$$

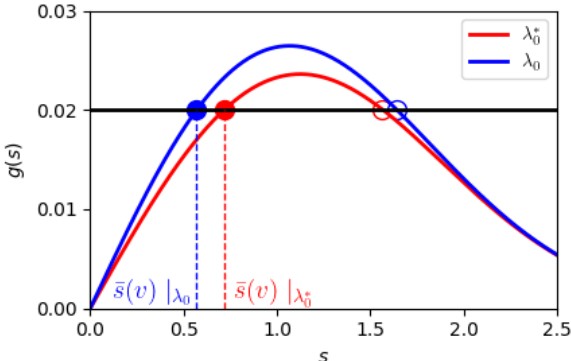

Figure S2: When $\beta < 0$, we have $\lambda_0 < \lambda_0^*$ and the coupled network dynamics thus have smaller tracking lag and larger maximal tractable speed. Filled circles: stable fixed points. Empty circles: unstable fixed points.

The lag distance $s$ and the maximal trackable speed $v_{\max}$ can be investigated by plotting $g(s)$ vs. $s$ (Figure S2). The function $g(s)$ is concave with respect to $s$, so there is one stable fixed point (on the left) and one unstable fixed point (on the right) when $g(s) = v$ (for $v < v_{\max}$). The stable fixed point on the $s$-axis is denoted as $\bar{s}(v)$, which represents the distance that the activity bump lags behind the stimulus.

Similar to the analysis in Section 2.3.2, we find that when $\beta < 0$, the lag distance $\bar{s}(v)$ is always smaller with E-INN dynamics compared to without E-INN dynamics, as expressed by the inequality

$$\bar{s}(v)\Big|_{\lambda_0} < \bar{s}(v)\Big|_{\lambda_0^*}$$

Furthermore, the maximal trackable speed $v_{\max}$ is given by $\max g(s)$. If the external stimulus moves faster than $v_{\max}$, the network cannot track it. In the case of $\beta < 0$, we have

$$\max_{\lambda_0} g(s) > \max_{\lambda_0^*} g(s).$$

implying that the CANN dynamics with E-INN dynamics can tolerate a higher maximal speed compared to without E-INN dynamics.