# OpenReview forum: "Slow and Weak Attractor Computation Embedded in Fast and Strong E-I Balanced Neural Dynamics"
_NeurIPS.cc/2023/Conference — NeurIPS 2023 spotlight_

### Official Review · Reviewer_Mp8Q · 2023-06-16

**Soundness:** 4 excellent
**Presentation:** 3 good
**Contribution:** 3 good
**Rating:** 7
**Confidence:** 3

**Summary:**

The paper combines two established models in theoretical neuroscience, the continuous attractor neural network (CANN), and the excitation-inhibition balanced neural networks (E-INN). Though those models have been applied to explain various phenomena in the brain, they possess seemingly contradictory characteristics, begging the question of how they can coexist in the brain, which is what this paper is about.
More specifically, the paper first introduces a plausible spiking model with three neuron populations that incorporates both CANN and E-INN.
Then, this spiking model is numerically simulated and shown to not only preserve the known properties of both CANN and E-INN, but also to benefit synergistically from one another through sped-up convergence, faster adaptation to change, and smaller tracking lag for varying stimuli.
Finally, the paper introduces a firing-rate model to account for the benefits observed in simulations. The main insight is obtained with the derivation of the eigenvalue of the dominant motion mode as a function of the $\beta$ coefficient, which accounts for the presence of E-I balance.

**Strengths:**

The strengths of the paper are:

- It is well-written and easy to follow despite being theory-heavy. It is also clearly structured.

- It tackles the important question of how to reconcile E-I balance with attractor models, and the presented insights might transfer to other attractor models than CANNs.

- It is a creative combination of existing ideas.

**Weaknesses:**

The current weaknesses of the paper are:

- Many equations in the spiking model are not homogeneous, so some constant might be missing. E.g. in Eq 1 a voltage (left) is equal to a current (right). In Eq 4, $f_{j}^{b}$ should be a voltage but it is equal to the inverse of a time. Same for Eq. 7. The paper would gain in clarity if the units are correctly worked out, although this does not change the message of the paper.

- The body text does not seem self-contained, I could only find the definitions of some quantities such as $U_E$, $r_E$, $k$ and $[\cdot]_{+}$ (line 202) in the appendix.

**Questions:**

Suggestions:

- Figure 1 (F) caption could be more specific about what quantity is the eigenvalue from, since the answer comes only in section 5.

- The shades in Figs 3 (A) Bottom, 5 (A,B,C) are not defined in the caption. Same for error bars in Fig 4 (D).

Questions:

- For the spiking model, how is the equilibrium of the CANN part defined theoretically? Is there a way to define an energy function?

- The CANN requires symmetric connectivity to function as an attractor network, which is often not considered a plausible connectivity motif. How can this be accounted for in the model?

- I don't see from Fig 4 (D) bottom how one can conclude "Shunting inhibition strength is proportional to the total EPSCs.". It feels like another type of plot would better demonstrate proportionality. Could you please explain in more details?

**Limitations:**

The authors do not mention any limitations of their model.

---

> ### Author Rebuttal · Authors · 2023-08-04
>
> Thanks for the encouraging and valuable comments of the reviewer, which are very helpful for us to improve the paper. Below, we would like to address reviewer’s concerns in weakness and questions in detail.
>
> **On weakness:**
>
> Thank the reviewer for pointing out that there exist inconsistencies of units in some equations in our paper. We will correct all these inconsistencies in the revision. Also, we will add the missing definitions for some quantities in the main text as suggested by the reviewer.
>
> **On questions:**
>
> 1.	The eigenvalues in Fig. 1F refer to the eigenvalues of CANN dynamics projected onto its dominating motion modes. We will modify the caption of Fig.1F to describe this clearly in the revision.
>
>
> 2.	Shades in Fig. 3A are the instantaneous readout results (computed at each dt=0.01ms), and the solid line is the running average calculated over 150 dts (i.e. 1.5ms). Shades in Fig. 5 indicate std. calculated over 20 trials. Error bars in Fig. 4D are std. calculated over 10 trials. We will add these details in figure captions in the revised manuscript.
>
>
> 3.	In the field, the equilibrium of a spiking CANN has never been theoretically defined, and people normally verify an equilibrium state by simulation. For a rate-based CANN, it can define an energy function if the input-output function is local. But for the rate-based CANN model we consider in Section 5 and Supplementary Material, since global divisive normalization (corresponding to shunting inhibition) is used, it does not have an energy function, although its equilibrium state can be analytically solved.
>
>
> 4.	Indeed, all CANN models consider symmetrical connections, which is not fully biologically plausible. In reality, this assumption can be regarded as a good approximation in many cases, since CANNs have successfully modelled many neural system behaviors.
>
>
> 5.	Fig 4D shows shunting inhibition (orange bar) is the strongest for the center neuron when the stimulus is positioned at the center, and the weakest for peripheral neurons. Indeed, this plot does not directly show *proportionality*. One has to refer to Equation (6) to see this where we define shunting inhibition as a product of total EPSCs and IPSCs from PV-expressing neurons. We will refine this statement in the revised manuscript.

---

> > ### Comment · Reviewer_Mp8Q · 2023-08-11
> >
> > Thanks to the author for addressing my remarks. I have no further comments and still recommend acceptance.

---

### Official Review · Reviewer_bo6j · 2023-07-05

**Soundness:** 3 good
**Presentation:** 2 fair
**Contribution:** 3 good
**Rating:** 7
**Confidence:** 3

**Summary:**

The authors present a model of a continuous attractor network in a balanced-state spiking network.
The model combines fast and slow connections. The fast connections are in charge of the balanced state and are thus strong 1/sqrt(K). The slow connections are in charge of the continuous attractor, and are weak 1/K. The model also has shunting inhibition which helps keep the network in the balance regime.
The resulting model has irregular activity, along with bump dynamics. The balanced state helps the network respond faster to changing inputs. The authors also provide an approximate firing rate model that gives intuition on the network dynamics.


**Strengths:**

The authors combine two categories of models that are often studied separately, and demonstrate their harmonious coexistence.
The analytical approximation of the interaction between the two components is very helpful and could be generalized to other models as well.



**Weaknesses:**

The stability of the bump in figure 3A bottom – it seems that the bump decays after stimulus offset.
I am not an expert on balanced networks, so I hope I didn’t miss something crucial. The authors claim that localized input disrupts balance in unstructured networks (line 137-138). The work of Hansel and van Vreeswijk 2012 shows that tuned input does not disrupt balance.
Another claim (line 40) is that E-I balance requires highly unstructured connectivity. This seems at odds with the work of Darshan et al (PRX 2018).


**Questions:**

Figure 2 shows some properties of balanced networks, but not two that are often used in the literature: Fano factor and the log-normal distribution of firing rates.
Table S4, S5: Why these numbers? Were they tuned somehow? Is the model robust to this choice?
Table S6, is I_d zero?
Line 98: cluster / clutter
Ref 17: missing bibliographic info
Supporting, line 11. Scaled by 1/K (not K) ?
Supporting, equation above line 4: sqrt(N) or sqrt(K)?
Line 101. Should the fast inhibitory be slow inhibitory?


**Limitations:**

yes

---

> ### Author Rebuttal · Authors · 2023-08-04
>
> Thanks for the encouraging and valuable comments of the reviewer, which are very helpful for us to improve the paper. Below, we would like to address reviewer’s concerns in weakness and questions in detail.
>
>
> 1.	Localized inputs disrupt the balance condition, if the EI balance dynamics is mediated by global unstructured connections. To mitigate this problem, several modelling studies (e.g., Rosenbaum and Doiron, 2014; Rosenbaum et al., 2017 and also the two studies the reviewer mentioned) use local connectivity to construct balance networks. These works have not explored the co-existence of EI balance and CANN. Notably, these modelling works do not refute our model assumption. The assumption in our model **explicitly** requires that the observed local connectivity structure come from the small synapse set, while the EI balance dynamics is still mediated via global unstructured connections. While an EI balance network with local connectivity could work when faced with localized inputs as in previous literature, our hypothesis is more aligned with the recent experimental data in (Scholl et al., 2021 Nature), where in Fig2a of the paper, the data shows null relationships of large connection weights between neuronal selectivity and orientation preference difference.
>
> 2.	Thanks for the suggestion. We plot the Fano factor of neuronal responses in the appended PDF, which is around $1$, indicating irregular neuronal responses in our model satisfies the Possion statictics. We will replace Fig.2C with it.
>
> 3.	We set $I_d$ as zero because in the brain, inhibitory neurons usually serve as interneurons and do not receive feedforward inputs from other cortical areas. Nonetheless, $I_d$ can be non-negative and it would not affect our results.
>
> 4.	The inhibition in our model is fast for both EI balance and CANN dynamics. The reasons for us using fast inhibition in CANN are two-fold: 1) slow negative feedback is unstable for a dynamical system and would easily lead to oscillations which could severely limit our choice of parameters; 2) the rate model requires fast inhibition so we can absorb the effect of $I_p$ into Equation (11) which permits an analytical solution. Notably, this setting is also biologically plausible, as the GABA dyanmics is much slower than the NMDA dynamics involved in the CANN.
>
> 5.	During our investigation, we first determined the parameters for EI balance dynamics (i.e. Table S4, S6). The classical EI balance constraint $\frac{f_E}{f_I}>\frac{w_{E I}}{w_{I I}}>\frac{w_{E E}}{w_{I E}}$ is only a necessary condition, and we found some parameters satisfying this constraint can still lead to oscillatory activity on a population level. We chose the values in Table S4 and S6 simply because they give nice irregular activity patterns. There is a pretty large parameter space that satisfies this requirement. We then determine the parameters for CANN dynamics. The most important parameter is $w_{max}^{EE}$, i.e., the recurrent connection strength in CANN dynamics. It cannot be too large, which would lead to recurrent dynamics stronger than EI balance dynamics and thus defeat the purpose of this study. It cannot be too small as well, as otherwise the plateau activity would decay very fast (but see also Fig. R2 in the appended PDF where we show increasing $w_{max}^{EE}$ by 25% gives persistent activity). Other parameters in Table S5 are basically chosen as we see fit. We are sure there is a large parameter space that achieves the same performance.
>
> 6.	We will also correct the typos the reviewer mentioned.

---

> > ### Comment · Reviewer_bo6j · 2023-08-13
> > **update**
> >
> > Thank you for the answers and clarifications.
> > I have increased my score to Accept.

---

### Official Review · Reviewer_FMpH · 2023-07-05

**Soundness:** 2 fair
**Presentation:** 3 good
**Contribution:** 3 good
**Rating:** 7
**Confidence:** 4

**Summary:**

The paper proposes a new spiking neural network where structured connectivity, consistent with a CANN, is combined with random connectivity, consistent with a excitation and inhibition network, with two inhibitory subpopulations. The network depends onj two different sets of weights: weak weights for CANN dynamics and strong weights for E-INN dynamics.
The network has better performance on a signal tracking task compared to a CANN network because of faster convergence of attractor states.

**Strengths:**

The proposed network that the paper discusses is the first to combine CANNs with balanced excitatory-inhibitory spiking neurons. The proposed network has some desirable properties in terms of ability to quickly track an input.
Furthermore, CANN dynamics with E-INN dynamics can tolerate a higher maximal speed for tracking compared to without E-INN dynamics.

**Weaknesses:**

It seems that the network does not truly have the same continuum of fixed points if $\beta\neq0$ as the CANN network would have.
This can be also seen with the diffusion of the "persistent activity" in Figure 3 after the stimulus is off. The CANN would have an actual persistent activity.
What is the trade-off how the proposed network might be faster, but does not have persistent activity (e.g. in terms of $\beta$)?

The CANN should also have a high readout after the stimulus is turned off because of the persistent activity.

The experiments  seems to be in contrast with the observation that the coupled network dynamics is neutrally stable in the second motion mode of the QHO.


As mentioned in 57, a total synaptic current of $\mathcal{O}(1)$ serves as input to the CANN network. This would actually shift the second motion mode of the QHO, making the "persistent" state not persistent.


The theory for this network suggests that an infinite number of neurons are needed to build a CANN network. However, this is physically impossible in the brain. It is possible to build a CANN network with a finite number of neurons [1]. A comparison of the tracking of a signal with such a network would benefit the analysis.



Overall, the methods used for the analysis are not novel.


[1] Noorman, Marcella, et al. "Accurate angular integration with only a handful of neurons." bioRxiv (2022): 2022-05.


**Questions:**

131: How do you justify that the final conclusions of our model are qualitatively applicable to general cases where $\beta$ is large, except for simulations?
The solution of the stationary state $\bar U_E$ in Equation S11 and through that $G^J (x; x0 j z)$ would be different.
Show that this change does not affect the relation $\lambda_0<\lambda_*$.

Did you perform simulations with a range of all the parameters?

Does it only hold for the particular set of parameter that is shown there?

When $\beta$ is sufficiently small, the stable solution is minimally affected by the E-INN.
How small does it need to be?

254: Does the CANN have a different equilibrium than the actual perfectly tracked signal? Would the equilibrium that is reached quickly for a sufficiently strong input not be the perfectly tracked signal?

Why are and two different inhibitory groups necessary for the network?

**Limitations:**

They are adequately discussed.

---

> ### Author Rebuttal · Authors · 2023-08-06
>
> Thanks for the encouraging and valuable comments of the reviewer, which are very helpful for us to improve the paper. Below, we would like to address reviewer’s concerns in weakness and questions in detail.
>
> **On weakness:**
>
> 1.	Whether the model can maintain a persistent bump is mainly dependent on the strength of the recurrent connection strength $w_{max}^{EE}$ of the CANN dynamics. In reality, the brain does not need to hold forever-lasting activity bumps. The computationally meaningful parameter region is that the network holds not permanent, but slow-decaying bump states, the so-called slow point dynamics [1]. Therefore, the parameters we choose in the main text is in this slow point dynamics region. Nonetheless, in Fig. R2 of the appended PDF, we demonstrate that by increasing the $w_{max}^{EE}$ 25%, the network can maintain the persistent state.
>
> [1] Sussillo, D., & Barak, O. (2013). Opening the black box: low-dimensional dynamics in high-dimensional recurrent neural networks. Neural computation, 25(3), 626-649.
>
>
> 2.	Indeed, the removal of the stimulus would result in a sudden drop in readout, but the dropped bump can last for a long or infinite time depending on the parameter setting.
>
>
> 3.	In Fig. R2, we demonstrate that the neutral stability holds for stronger recurrent connections of CANN dynamics. For the parameters we used in the main text, we can approximate the slow-decaying states as stable states, and hence the neutral stability in the second motion mode roughly holds.
>
>
> 4.	We presume the reviewer refers to that the total synaptic from EI balance dynamics shifts the _first_ motion mode (i.e., the bump height mode) and causes instability of the bump in the height direction. But we note that for $\beta< 0$, the network can still hold static bump if the CANN weights are large enough. However, we think computational meaningful states of CANN dynamics are the slow-decaying states as discussed above.
>
>
> 5.	In theoretical analysis, we assume that there are infinite number of neurons. To confirm the theoretical results, we often carry out simulations with a finite number of neurons, e.g., in our simulation, we chose $N=100$. The reviewer suggested an interesting reference. This paper proposes an interesting method to smoothen the energy barrier. As a side note, we also observed less discrete attractor space for $\beta<0$ in our model when CANN connection strength is inhomogeneous (not shown in this paper). We will include this paper in the related work section.
>
>
> **On questions:**
>
> 6.	Thanks for the insightful comments of the reviewer. It is true that to carry out theoretical analysis, we need small $\beta$ (we use small/large here to represent the magnitude of $\beta$). For large $\beta$, we can only validate the results by simulation. Fig.S1 show that if $\beta$ is not too large, $\lambda_0^\beta < \lambda_0^*$ still holds. However, there is no theoretical guarantee that $\lambda_0^\beta < \lambda_0^*$ always holds for arbitrarily large $\beta$. We will modify our statement in the revised manuscript.
>
>
> 7.	We performed simulations over a wide range of parameters. There is a very large parameter space in which our conclusion holds.
>
>
> 8.	The model holds for a relatively wide range of parameters, for example, for theoretical analysis, we require the EI balance weights scale in $\mathcal{O}(1/K)$ while the CANN weights in $\mathcal{O}(1/\sqrt{K})$. In practice, however, this scaling relationship can be largely relaxed, as long as the EI balance weights are much stronger than the CANN weights, our main results hold.
>
>
> 9.	To keep the bump stable, it needs $\beta$ to be smaller than a threshold, whose value depends on other parameters, such as the recurrent connection strength $J_0$ and the global inhibition strength $\kappa_c$. However, it is difficult to analytically calculate this threshold
>
>
> 10.	Sorry, we do not understand what the reviewer means by “the perfectly tracked signal”.
>
>
> 11.	Two different inhibitory groups are necessary, since they are needed to separate the two dynamics running on different time scales.

---

> > ### Comment · Reviewer_FMpH · 2023-08-15
> >
> > I thank the authors for their answers and clarifications. I have no further comments and still recommend acceptance.
> >
> >
> > 10. A perfectly tracked signal would one without any lag or mismatch.

---

### Official Review · Reviewer_kktH · 2023-07-11

**Soundness:** 3 good
**Presentation:** 2 fair
**Contribution:** 2 fair
**Rating:** 7
**Confidence:** 3

**Summary:**

This study explores the compatibility of attractor networks and excitation-inhibition balanced networks (E-INNs) in neural circuits. It proposes that a neural circuit can exhibit traits of both by utilizing two sets of synapses: one set for strong and fast irregular firing and another set for weak and slow attractor dynamics. Simulations and analysis show that this approach enhances network performance, including accelerated convergence of attractor states and preserved E-I balanced conditions.

**Strengths:**

* The approach addresses the challenge of reconciling the structural demands of attractor networks and E-INNs, which are typically studied independently. By combining two sets of synapses with different properties, the proposed approach allows for the coexistence of both attractor dynamics and irregular firing in a neural circuit.

* The simulations and theoretical analysis demonstrate that the approach leads to improved network performance compared to using only one set of synapses. The enhanced performance includes accelerated convergence of attractor states and the retention of E-I balanced conditions for localized input. This suggests that the combined approach can achieve the advantages of both attractor networks and E-INNs simultaneously.

* The study provides insight into how structured neural computations can be realized through the integration of irregular firings of neurons. By investigating the coexistence of attractor networks and E-INNs, the approach sheds light on the mechanisms underlying complex neural processing, contributing to a better understanding of neural computation in the brain.

**Weaknesses:**

* The authors do not discuss the biological plausibility of the proposed two-set synapse model. It is essential to consider whether such a system could be implemented in actual neural circuits and whether it aligns with known biological mechanisms.

*  Another drawback of the paper is the lack of proper discussion of how general these findings are when the paper's assumptions are softened or when dealing with sufficiently different learning algorithms. When studying biological featuers in silico, qualitative and quantiative robustness is extremely important.

**Questions:**

See above.

---

> ### Author Rebuttal · Authors · 2023-08-04
>
> Thanks for the encouraging and valuable comments of the reviewer. Below are our replies to reviewer’s concerns.
>
> 1.	On the biological plausibility of our model. We actually just found an experimental study [1] (which we did not know when writing this paper), which has already provided strong evidence for the two-set synapse assumption in our model. In the experimental data, the authors measured synapse strengths using combined two-photon and scanning electron microscopy techniques. They found that **“no evidence that strong synapses have a predominant role in the selectivity of cortical neuron responses”**, which correspond to the unstructured and strong synapses for EI balance dynamics in our model. They also found that **“spatial clustering of co-active inputs appears to be reserved for weaker synapses”**, which corresponds to the weak synapses for CANN dynamics in our model. Please see the experimental figure in the appended PDF.
>
> 2.	On the generality of our model. The results of our model are quite general. There are no specific assumptions on the model, other than a requirement on the relationship between two set of synaptic weights: one scales in $\mathcal{O}(1/K)$ and the other in $\mathcal{O}(1/\sqrt{K})$, with K the connectivity of neurons. However, this strict scaling relationship is mainly for the convenience of theoretical analysis, as done in [2]. In practice, this scaling relationship can be largely relaxed, as long as the synaptic weights for EI balance dynamics are much stronger than those for CANN dynamics. For other parameters, they are all consistent with the normal parameter settings in EI balance and CANN dynamics. In the revised manuscript, we will discuss about the generality of our model.
>
> [1] Scholl, B., Thomas, C. I., Ryan, M. A., Kamasawa, N., & Fitzpatrick, D. (2021). Cortical response selectivity derives from strength in numbers of synapses. Nature, 590(7844), 111-114.
>
> [2] van Vreeswijk, C., & Sompolinsky, H. (1998). Chaotic balanced state in a model of cortical circuits. Neural computation, 10(6), 1321-1371.

---

> > ### Comment · Reviewer_kktH · 2023-08-16
> >
> > Following the rebuttal I have adjusted my score.

---

### Author Rebuttal · Authors · 2023-08-04

**On the biological plausibility of our model**

We are delighted to find that an experimental study [1], which we did not know when writing this NeurIPS paper, has already provided strong evidence for the key assumption of our model, namely, the neural circuit consists of two sets of synapses: a weak one for attractor computation and a strong one for EI balance dynamics. In this experimental work, the authors measured synapse strengths using combined two-photon and scanning electron microscopy techniques. They found that **“no evidence that strong synapses have a predominant role in the selectivity of cortical neuron responses”**, which corresponds to the strong and unstructured synapses for EI balance dynamics in our model. They also found that **“spatial clustering of co-active inputs appears to be reserved for weaker synapses”**, which indicates that the set of weak synapses determines the orientation selectivity of neurons (note that orientation selectivity is often modeled by a CANN [2]). Please refer to the experimental figure in the appended PDF.

[1] Scholl, B., Thomas, C. I., Ryan, M. A., Kamasawa, N., & Fitzpatrick, D. (2021). Cortical response selectivity derives from strength in numbers of synapses. Nature, 590 (7844), 111-114.

[2] Ben-Yishai, R., Bar-Or, R. L., & Sompolinsky, H. (1995). Theory of orientation tuning in visual cortex. Proceedings of the National Academy of Sciences, 92 (9), 3844-3848.

---

### Decision · Program_Chairs · 2023-09-21

**Decision:**

Accept (spotlight)

**Comment:**

This paper introduces a balanced spiking neural network that implements an approximate, continuous bump attractor network. It incorporates the advantages of fast response to input and slow decay of the bump using a biologically plausible synaptic architecture. Reviewers are enthusiastic about this technically sound paper.